



# Uncertainties and their interaction in flood hazard assessment with climate change

**Hadush Meresa, Conor Murphy, Rowan Fealy, and Saeed Golian**

Irish Climate Analysis and Research UnitS (ICARUS), Department of Geography, Maynooth University, Maynooth, Ireland

**Correspondence:** Hadush Meresa (hadush.meresa@mu.ie)

**Abstract.** The assessment of future impacts of climate change is associated with a cascade of uncertainty linked to the modelling chain employed in assessing local-scale changes. Understanding and quantifying this cascade is essential for developing effective adaptation actions. We evaluate and quantify uncertainties in future flood quantiles associated with climate change for four catchments, incorporating within our modelling chain uncertainties associated with 12 global climate models contained in the Coupled Model Intercomparison Project Phase 6, five different bias correction approaches, hydrological model parameter uncertainty and the use of three different extreme value distributions for flood frequency analysis. Results indicate increased flood hazard in all catchments for different Shared Socioeconomic Pathways (SSPs), with changes in flooding consistent with changes in annual maximum precipitation. We use additive chains and analysis of variance (ANOVA) to quantify and decompose uncertainties and their interactions in estimating selected flood quantiles for each catchment. We find that not only do the contributions of different sources of uncertainty vary by catchment, but that the dominant sources of uncertainty can be very different on a catchment-by-catchment basis. While uncertainties in future projections are widely assumed to be dominated by the ensemble of climate models used, we find that in one of our catchments uncertainties associated with bias correction methods dominate, while in another the uncertainty associated with the use of different extreme value distributions outweighs the uncertainty associated with the ensemble of climate models. These findings highlight the inability to generalise a priori about the importance of different components of the cascade of uncertainty in future flood hazard at the catchment scale. Moreover, we find that the interaction of components of the modelling chain employed are substantial (> 20 % of overall uncertainty in two catchments). While our sample is small, there is evidence that the dominant components of the cascade of uncertainty may be linked to catchment characteristics and rainfall–runoff processes. Future work that seeks to further explore the characteristics of the uncertainty cascade as they relate to catchment characteristics may provide insight into a priori identifying the key components of modelling chains to be targeted in climate change impact assessments.

## 1 Introduction

Climate change is likely to increasingly affect hydrological regimes and flood hazards over coming decades. Considerable changes in atmospheric temperature, precipitation, humidity and circulation are expected, which may result in increasing extreme events, including floods (IPCC, 2013). According to Rojas et al. (2013), flood frequency in Europe will increase due to climate change, with significant socio-economic implications for the region. Blöschl et al. (2017, 2019) conclude that the timing and magnitude of European floods have shifted due to climate change, and its consequences are not uniform across the region, with north-western Europe experiencing earlier and higher flood peaks. Modelling and understanding of catchment-scale flood hazard projections is therefore an important endeavour for informing adaptation strategies.

Climate change impact assessment is subject to considerable uncertainties (Wilby and Dessai, 2010; Smith et al., 2018); Blöschl et al. (2019) recently highlighted uncertainty in hydrology as one of the 25 challenges in hydrological science. Traditional or top-down climate change impact as-

sessments typically follow a modelling chain where output from global climate models (GCMs), forced with scenarios of future greenhouse gas concentrations, are extracted and scaled to represent a study catchment. Hydrological models, calibrated for current conditions, are then forced with these GCM outputs to create discharge series spanning multiple decades into the future. When assessing flood hazard, extreme value distributions are typically fitted to samples of extreme events (e.g. annual maximum flood series) representing current and future climates to evaluate changes in the characteristics of flooding. This modelling chain is replete with uncertainties that propagate and interact, resulting in potential large ranges in projected change at the catchment scale (e.g. Meresa and Gatachew, 2019) that can impede decision-making (Smith et al., 2018).

Numerous studies have attempted to quantify the uncertainties in future climate change impacts due to the use of different climate models (e.g Knutti and Sedláček, 2013), natural variability (e.g. Giuntoli et al., 2018; Hughes et al., 2011), bias correction techniques (e.g. Kay et al., 2009; Saini et al., 2015; Soriano et al., 2019), downscaling approaches (e.g. Fowler et al., 2007; Gutmann et al., 2014), hydrological modelling uncertainties (e.g. Wilby and Harris, 2006; Bastola et al., 2011a; Addor et al., 2014; Meresa and Romanowicz, 2017; Broderick et al., 2019) and the application of different extreme value distributions for flood hazard estimation (e.g. Meresa and Romanowicz, 2017; Lawrence, 2020). Findings from such studies confirm that the dominant source of uncertainty varies across studies. For example, Jobst et al. (2018) concluded that the GCM structure is the dominant contributor followed by emission scenario, bias correction and hydrological model structure in New Zealand catchments. Peleg et al. (2015) found that the hydrological regimes are highly sensitive to changes in convective precipitation in eastern Mediterranean catchments. Gosling et al. (2011) evaluated seasonal flow changes in UK catchments and found greater uncertainty in future estimates due to climate models than hydrological models. In Norwegian catchments, Lawrence (2020) highlighted that climate models and frequency models are dominant sources of uncertainty in future flood projections. Therefore, the key components of the uncertainty cascade and their contributions tend to vary depending on the region, precipitation type and catchment.

Recognition of the uncertainties inherent in future climate change impacts has also given rise to novel approaches to decision-making that embrace uncertainties (e.g. Wilby and Murphy, 2019; Clark et al., 2016). Rather than aiming to derive precise assessments of future risk, such approaches aim to stress test and evaluate adaptation options, either in static or dynamic ways (Mazzorana et al., 2012), to identify actions that are functional across a range of plausible future conditions rather than optimised to a certain outcome (e.g. structured decision-making (SDM), robust decision-making, decision scaling and adaptation pathways). Shepherd et al. (2018) recommend a storyline approach for navigating uncertainty,

whereby examination of recent, notable extremes and the development of plausible narratives of change in their drivers enables a novel assessment of future changes, placing emphasis on physical processes. Each of these approaches requires the cascade of uncertainty to be evaluated or navigated in ways that can better inform adaptation. For example, Broderick et al. (2019) employ a scenario neutral framework (Prudhomme et al., 2011) to evaluate design allowances for flood defences, taking into account uncertainties derived from a large climate model ensemble, natural variability and hydrological models.

Whether employing traditional impact-led or novel decision-centric approaches, it is critical that key components of the cascade of uncertainty are adequately included. Research over the past 2 decades clearly shows that methodological choices made in assessing future climate change fundamentally impact on the portrayal of climate risk (Clark et al., 2016; Melsen et al., 2019). Top-down approaches must a priori decide on the components of the modelling chain to include. Including all relevant components is typically a very resource expensive task (Smith et al., 2018), and it may not be apparent at the outset which components of the uncertainty cascade to include for specific catchments. For bottom-up approaches, such as the scenario-neutral method (Broderick et al., 2019), the modeller is forced to choose which aspects of uncertainty to include in assessing sensitivity to future changes. Again, this may not be apparent from the outset. It is widely acknowledged that climate models exhibit biases in their outputs (Krinner and Flanner, 2018; Giorgi and Gao, 2018), with numerous studies highlighting the need to post-process climate model output before use in simulating hydrological response (Ehret et al., 2012; Teng et al., 2012; Osuch et al., 2017; Meresa and Romanowicz, 2017). Dobler et al. (2012) suggest that the dominant components of the modelling chain may further vary, depending on which part of the flow regime (low, mean and high flows) is of interest. They highlight, for example, that the importance of bias correction approaches to the overall uncertainty in hydrological response increases for high flows. However, there is still no consensus on which bias correction techniques are most effective, nor in how bias correction techniques can modify future climate change signals. For instance, Teutschbein and Seibert (2013) and Yang et al. (2010) showed distribution mapping based on theoretical distributions outperforms other bias correction methods. Similarly, Berg et al. (2012) and Chen et al. (2013) show that theoretical distribution mapping performs similar to, or marginally better than, empirical quantile mapping. On the contrary, Gudmundsson et al. (2012), Gutjahr and Heinemann (2013) and Lafon et al. (2013) show that empirical quantile mapping demonstrates higher skill than theoretical distribution mapping in systematically correcting precipitation.

Moreover, deep uncertainty can arise by virtue of the ad hoc ways in which components of possible modelling chains are assembled and characterised within the overall modelling

framework adopted (Wilby and Murphy, 2019). Yet there are currently no established ways of sampling from the hierarchy of models used to evaluate impacts of climate change (Clark et al., 2016). Therefore, an important step in better integrating future climate risk into decision-making is quantifying and partitioning the contribution of different components of uncertainty and their interactions to help scientists and decision-makers better navigate the cascade of uncertainty by identifying contributing sources that should be more fully explored (Smith et al., 2018). In this regard, ANOVA (ANalysis Of VAriance)-based techniques, which can be used to decompose sources of uncertainty and their interaction, offer considerable utility (e.g. Kay et al., 2009; Vetter et al., 2016; Hattermann et al., 2018; Meresa, 2020). In this study, we explore uncertainties in future flood hazard for four catchments, representative of different flood response types, in Ireland and employ additive chains and variance decomposition to quantify and examine the contribution of various sources of uncertainty, together with their interaction, to the overall uncertainty in flood hazard projections. In doing so, we place emphasis on evaluating the uncertainties derived from (i) climate models in the newly available Coupled Model Intercomparison Project Phase 6 (CMIP6) ensemble (Wyser et al., 2020), (ii) widely used bias correction techniques, (iii) hydrological model parameter uncertainty and iv) the use of different extreme value distributions. The remainder of the paper is organised as follows: Sect. 2 outlines the study design and data/methods employed, and Sect. 3 presents key results, exploring uncertainties for key steps in the modelling chain examined, together with their contributions and interactions to the full range of projected change in flood risk. Section 4 provides a discussion of key insights, limitations and future directions before drawing main conclusions in Sect. 5.

## 2 Models and study design

Our study design is illustrated in Fig. 1. We quantify uncertainties and their interaction in projected flood hazards using 12 climate models (CMs) contained in the Coupled Model Intercomparison Project Phase 6 (CMIP6) ensemble (https://esgf-node.llnl.gov/search/cmip6/, last access: 6 June 2020), forced using three Shared Socioeconomic Pathways (SSPs) scenarios. We use five bias correction techniques (change factor – CF; double gamma distribution quantile – DGQM; Birnbaum distribution quantile mapping – BQM; single gamma distribution quantile – SGQM; empirical quantile – EQM) to post-process climate model outputs (daily precipitation and air temperature time series). The daily bias-corrected precipitation and temperature data are used as input to the Génie Rural à 4 paramètres Journalier (GR4J) hydrological model (Perrin et al., 2003). The generalised likelihood uncertainty estimation (GLUE) technique is used to quantify GR4J parameter uncertainty. In assessing future flood hazard, the sensitivity of results to different extreme

value distributions is examined. In addition to evaluating each source of uncertainty independently, we use variance decomposition to quantify the contribution of each methodological choice and their interactions to the overall uncertainty in assessing flood risk. The climate change impact is evaluated based on relative future changes in the magnitude of floods for the 2020s (2010–2039), 2050s (2040–2069) and 2080s (2070–2099) with respect to the reference period (1976–2005). The following sections provide further details on the study catchments and each stage of the modelling chain employed.

### 2.1 Study catchments and hydro-climate data sets

Simulations are undertaken for four catchments (Boyne, Blackwater, Newport and Slaney), representing different flood response types across Ireland (Broderick et al., 2019). Their location, together with a summary of hydro-climatic conditions for each catchment, is summarised in Table 1 and Fig. 2. Catchment area ranges from $146\,km^2$ (Newport) to $2447\,km^2$ (Boyne), while mean elevation ranges from 56 m (Newport) to 112 m (Blackwater). For each catchment, we use gridded ($1 \times 1\,km$) daily precipitation and temperature data (Walsh, 2012) area averaged for the period 1976–2005 to provide a single representative baseline series for each catchment. Daily potential evapotranspiration is derived using air temperature and elevation, following the method of Hamon (1963). This approach is favoured over less parsimonious but more physically based methods (e.g. Penman–Monteith) which have greater data input requirements (e.g. wind speed and humidity) not available for all study catchments. Daily discharge data for each catchment were obtained from the Office of Public Works (OPW; http://www.epa.ie/hydronet/, last access: 10 February 2020).

### 2.2 Climate projections and bias correction

Daily precipitation (pr) and air temperature (tas) time series for the period 1971–2100 were extracted for 12 members of the CMIP6 ensemble (https://esgf-node.llnl.gov/search/cmip6/, last access: 6 June 2020) forced by each of the three SSP (SSP126, SSP370 and SSP585) scenarios (see Table 2 for details). For each catchment, daily precipitation and air temperature were extracted from the closest land-based CM grid overlying the catchment centroid. We employ five commonly used techniques to bias correct raw climate model output and to examine the contribution of the selected bias correction methods to the total uncertainty in future flood hazard.

#### 2.2.1 Change factor /delta change (CF)

The change factor technique applies multiplicative (for precipitation) and/or additive (for temperature) procedures for correcting raw model output. This involves correcting simulated daily precipitation ($P_{fut,corr}$) by multiplying the ra-

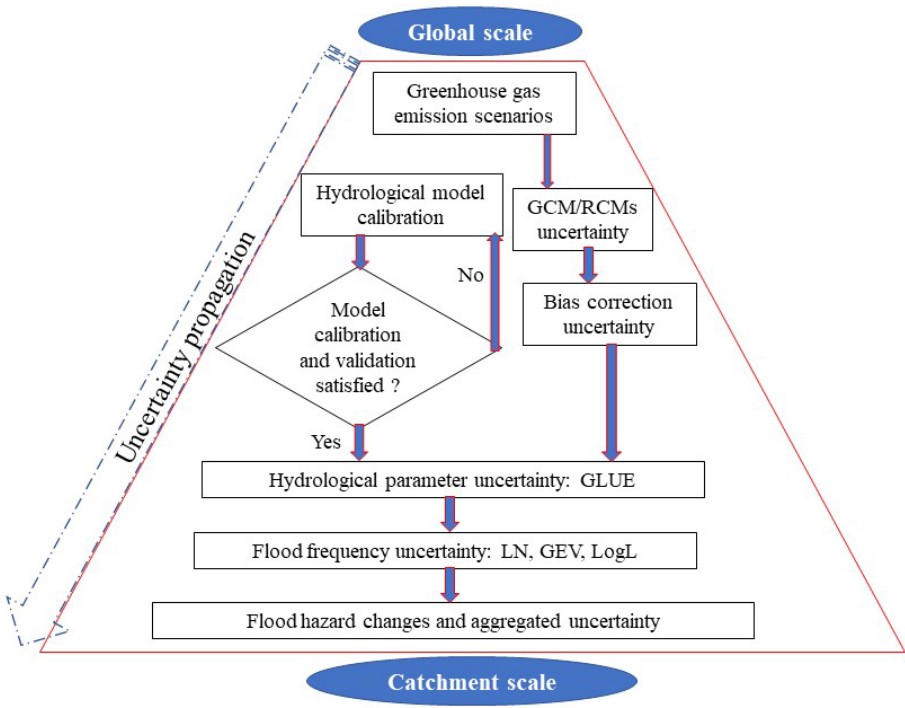

**Figure 1.** Research flow chart to estimate projections of flood hazard and identify associated uncertainty.

**Table 1.** Statistical and hydro-climatic characteristics of each study catchment.

|  | Boyne | Blackwater | Slaney | Newport |
|---|---|---|---|---|
| Latitude (degree) | 53.60 | 52.14 | 52.80 | 53.95 |
| Longitude (degree) | −6.96 | −8.94 | −6.61 | −9.44 |
| Catchment area (km$^2$) | 2447.36 | 1257.20 | 1032.68 | 146.02 |
| Mean elevation (m) | 70.00 | 116.00 | 107.00 | 56.00 |
| Mean of annual max precipitation (mm) | 28.58 | 39.17 | 38.23 | 41.69 |
| Mean of annual max streamflow (m$^3$/s) | 232.22 | 233.59 | 137.39 | 24.26 |
| Lower quantile of precipitation ($P85$) (mm) | 0.01 | 0.02 | 0.01 | 0.04 |
| Higher quantile of precipitation ($P15$) (mm) | 5.46 | 8.25 | 6.60 | 9.95 |
| Low flow ($Q95$) (m$^3$/s) | 4.67 | 4.47 | 4.10 | 0.81 |
| Peak flow ($Q5$) (m$^3$/s) | 106.22 | 105.40 | 53.30 | 16.20 |
| Coefficient of variance of streamflow (–) | 37.97 | 23.79 | 30.92 | 18.07 |
| Coefficient of variance of precipitation (–) | 29.34 | 24.68 | 30.86 | 33.17 |
| Base flow index (–) | 0.72 | 0.62 | 0.73 | 0.69 |
| Mean of surface runoff (m$^3$/s) | 10.21 | 13.14 | 5.77 | 1.86 |
| Mean of base flow (m$^3$/s) | 26.85 | 21.78 | 15.58 | 4.19 |

tio of observed precipitation ($P_{obs}$) and reference precipitation simulation ($P_{ref,raw}$) to future simulations of raw climate model precipitation ($P_{fur,raw}$). For correcting future air temperature ($T_{fut,corr}$), the difference in observed air temperature ($T_{obs}$) and simulated temperature in the reference period ($T_{ref,raw}$) is added to raw climate output ($T_{fur,raw}$).

$$P_{fut,corr} = P_{fur,raw} \cdot \frac{P_{obs}}{P_{ref,raw}} \tag{1}$$

$$T_{fut,corr} = T_{fur,raw} + (T_{obs} - T_{ref,raw}). \tag{2}$$

### 2.2.2 Empirical quantile mapping

Empirical quantile mapping (EQM) is based on pairwise comparison between the empirical cumulative density functions (ecdf) of observed and simulated daily precipitation time series during the reference period (1976–2005). This is a purely empirical approach with direct matching of the histogram of the observed precipitation to the future period. Future precipitation and temperature are corrected using the

**Table 2.** List of CMIP6 climate models employed in this study.

| Code | Institute | Parent source ID | Institution ID |
|------|-----------|------------------|----------------|
| CM1 | Commonwealth Scientific and Industrial Research Organisation, Australia | ACCESS-CM2 | CSIRO |
| CM2 | Beijing Climate Center, China | BCC-CSM2-MR | BCC |
| CM3 | National Center for Atmospheric Research, USA | CESM2 | NCAR |
| CM4 | European EC – Earth consortium | EC-Earth | EC-Earth consortium |
| CM5 | Global Fluid Dynamics Laboratory, USA | GFDL | NOAA-GFDL |
| CM6 | Met Office Hadley Centre, UK | HadGEM3-GC31-LL | MOHC |
| CM7 | JAMSTEC, AORI, NIES and R-CCS, Japan | MIROC6 | MIROC |
| CM8 | Max Planck Institute for Meteorology, Germany | MPI-ESM1-2-HR | MPI-M |
| CM9 | Meteorological Research Institute, Japan | MRI-ESM2-0 | MRI |
| CM10 | Nanjing University of Information Science and Technology, China | NESM3 | NUIST |
| CM11 | NorESM climate modelling consortium, Norway | NorESM2-LM | NCC |
| CM12 | Met Office Hadley Centre, UK | UKESM1-0-LL | MOHC |

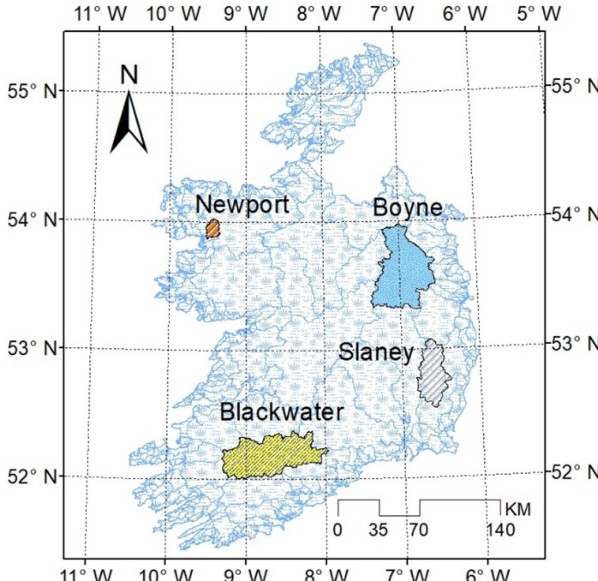

**Figure 2.** Location of the selected study catchments.

inverse of the ecdf (ecdf$^{-1}$) and fitted ecdf ecdf$_{\text{hst,m}}$.

$$P_{\text{hst,m}}^{\text{cor}} = (\text{ecdf}_{\text{obs,m}}^{-1}(\text{ecdf}_{\text{hst,m}}(P_{\text{hst,m}}))) \qquad (3)$$

$$T_{\text{hst,m}}^{\text{cor}} = (\text{ecdf}_{\text{obs,m}}^{-1}(\text{ecdf}_{\text{hst,m}}(T_{\text{hst,m}}))). \qquad (4)$$

The non-parametric quantile matching was performed first in the reference period (m – modelling period) using an exponential transfer function assumption. Then the calibrated coefficients were used to correct the future daily precipitation and air temperature.

### 2.2.3 Distribution quantile mapping

Distribution quantile mapping (DQM) is a distribution-parameter-dependant bias correction technique. The parameters are extracted by fitting gamma distribution to observed

and simulated time series data and matching its corresponding quantiles from the observed and raw climate model output in the reference period (1976–2005; Piani et al., 2010). We applied two types of DQM to correct climate biases, namely single gamma distribution quantile mapping (SGDQM) and double gamma distribution quantile mapping (DGDQM), together with the Birnbaum–Sanders distribution method (BSM; Osuch et al., 2016). These methods also allow for any excess in the number of dry, drizzle and wet days to be considered and corrected. In the case of SGDQM, the first step is fitting a gamma distribution to the upper 75 % of daily observed and raw climate output precipitation distribution. Whereas, in the DGDQM, the gamma distribution is fitted to both the upper $\geq 75\,\%$ and to the lower $< 75\,\%$ of the daily observed and raw climate output precipitation in the reference period (1976–2005). In both cases, non-rain days were removed, and only wet days were considered. Similarly, BSM used the Birnbaum–Sanders distribution to transfer the precipitation quantile from the observed time series to raw output of the CMs in the reference period.

$$P_{\text{corr}} = F_{\text{dg}}^{-1}(F_{\text{dg}}(P_{\text{raw}}(t), \alpha_{\text{raw}}, \beta_{\text{raw}}) \alpha_{\text{Obs}} \beta_{\text{Obs}}) \qquad (5)$$

$$P_{\text{corr}} = F_{\text{db}}^{-1}(F_{\text{db}}(P_{\text{raw}}(t), \alpha_{\text{raw}}, \beta_{\text{raw}}) \alpha_{\text{Obs}} \beta_{\text{Obs}}) \qquad (6)$$

$$T_{\text{corr}} = F_{\text{db}}^{-1}(F_{\text{db}}(T_{\text{raw}}(t), \alpha_{\text{raw}}, \beta_{\text{raw}}) \alpha_{\text{Obs}} \beta_{\text{Obs}}) \qquad (7)$$

$$T_{\text{corr}} = F_{\text{dn}}^{-1}(F_{\text{dn}}(T_{\text{raw}}(t), \alpha_{\text{raw}}, \beta_{\text{raw}}) \alpha_{\text{Obs}} \beta_{\text{Obs}}), \qquad (8)$$

where $P_{\text{corr}}$ and $T_{\text{corr}}$ are the bias-corrected daily precipitation and temperature, respectively. Likewise, $P_{\text{raw}}(t)$ and $T_{\text{raw}}(t)$ represent the raw climate model output for precipitation and temperature, respectively. The raw climate output inverse cumulative density (CDF) is symbolised by $F_{\text{dg}}^{-1}$, $F_{\text{db}}^{-1}$ and $F_{\text{dn}}^{-1}$ for precipitation and temperature, respectively. The dn, db and dg subscripts represent the normal (for temperature), Birnbaum–Sanders (for both precipitation and temperature) and gamma (for precipitation) distributions, respectively. The gamma (for precipitation) and Birnbaum–Sanders distributions have two parameters, the shape and scale parameters

which are symbolised by $\alpha$ and $\beta$, and the normal (for temperature) distribution, with mean and standard deviation represented by $\mu$ and $\sigma$, respectively.

## 2.3   Bias correction performance evaluation

The performance of the selected bias correction techniques was evaluated using the following four statistical measures: Pearson correlation (RR), mean absolute error (MAE), root mean square error (RMSE) and percent bias (PBIAS). Evaluation was performed by comparing the ability of each approach to capture observed precipitation and temperature.

$$RR = \frac{\sum (P_S - \overline{P_S}) \cdot (P_C - \overline{P_C})}{\sqrt{\sum (P_S - \overline{P_S})^2 \cdot \sum (P_C - \overline{P_C})^2}} \qquad (9)$$

$$MAE = \frac{\sum_{i=1}^{n} |P_S - P_C|}{N} \qquad (10)$$

$$PBIAS = \frac{\sum_{i=1}^{n} |P_S - P_C|}{\sum_{i=1}^{n} P_S} \qquad (11)$$

$$RMSE = [\frac{\sum_{i=1}^{n} |P_S - P_C|^2}{N}]^{0.5}, \qquad (12)$$

where $P_S$ and $P_C$ are observed and corrected precipitation, respectively, $\overline{P_C}$ is the mean of corrected precipitation, $\overline{P_S}$ is the mean of observed precipitation, and $N$ is the number of observations.

## 2.4   Hydrological modelling

The GR4J model (Perrin et al., 2003) is a parsimonious four parameter, lumped conceptual rainfall–runoff model that has been widely applied in different hydro-climate conditions (Meresa and Gatachew, 2019; Meresa et al., 2017; He et al., 2018). The model is particularly suited to climate change impact studies given its parsimonious structure. GR4J simulates streamflow using precipitation, temperature and potential evapotranspiration (Perrin et al., 2003) and has previously been successfully used in Ireland to assess climate change impacts (Broderick et al., 2016, 2019), river flow reconstructions (O'Connor et al., 2020) and seasonal hydrological forecasting (Donegan et al., 2021). Broderick et al. (2019) highlight the good performance of GR4J in capturing a range of hydrological signatures across diverse Irish catchments (Broderick et al., 2019). The model has two consecutive stores, i.e. one related to runoff production and the other to runoff routing. Detailed information about the model structure is given in Perrin et al. (2003). The upper and lower limits of the four model parameters are listed in Table S1 in the Supplement. The model was calibrated and validated using observations over the period 1976–2005. The first 4 years were used as a model warmup to stabilise the initial states of the hydrological parameters. Calibration was undertaken from 1981–1999 and validation from 2000–2005. The Nash–Sutcliffe efficiency (NSE) objective function was used to evaluate the model performance and to identify behavioural parameter sets (Nash and Sutcliffe, 1970). The NSE is defined as follows:

$$NSE = 1 - \frac{\sum_{t=1}^{j} (Q_{o,t} - Q_{m,t})^2}{\sum_{t=1}^{j} (Q_{o,t} - \overline{Q}_o)^2}, \qquad (13)$$

where $Q_{o,t}$ and $Q_{m,t}$ are observed and simulated flow at time $t$, $Q_o$ is the mean observed flow, and $j$ is the length of the $j$th time series. The ability of hydrological model simulations to replicate the observations was further evaluated by deriving the proportion of extreme high flows inside the 95 % confidence intervals (PCI). This is helpful for identifying a fixed NSE threshold for differentiating between behavioural and non-behavioural simulations (Li et al., 2011; Xu, 2014).

The generalised likelihood uncertainty estimation (GLUE) approach is widely applied to quantify hydrological model parameter uncertainty (Beven and Binley, 1992). GLUE is an informal statistical approach that uses Monte Carlo (MC) simulation to generate many possible hydrological parameter sets from specific ranges. The choice of the number of samples is somewhat subjective, with studies typically employing a few thousand to tens of thousands of hydrological model parameter set samples to simulate river flow (Zheng and Bennett, 2002). In applying GLUE, we randomly generated a large number of parameter sets (30 000) using a uniform distribution constrained by the acceptable ranges of each GR4J parameter. Behavioural parameter sets were identified based on NSE (Eq. 13) scores > 0.50.

## 2.5   Flood frequency analysis

Numerous extreme value distributions can be deployed to estimate the frequencies of high flows. For example, the log-Pearson III (LP3) distribution is very popular in the USA and Australia for infrastructure design (Griffis and Stedinger, 2007), the generalised extreme value (GEV) and Pearson type III distributions are widely used in Europe (Madsen et al., 2014), while the Wakeby and log-normal (LN) distributions have been frequently used in Asia (Chen et al., 2012). However, a single distribution model may not be able to capture the entire temporal and spatial variability of hydrological extremes. Therefore, we employ three common distribution types (LN, log-logistic and GEV) with maximum likelihood parameter estimator (MLE) for flood frequency curve development. MLE estimates the distribution parameters by optimising the likelihood function of the cumulative probability distribution density, and its reliability is evaluated using the standard error of the estimated parameters. These distributions were fitted to the annual maximum daily peak flows in each study catchment. In Eqs. (14)–(16), the respective probability density function (PDF) of each distribution is presented. The log-normal and log-logistic models have two pa-

rameters, while the GEV distribution has three parameters.

Log-Normal (LN) $f(x) = \dfrac{\exp\left(-\frac{1}{2}\left(\frac{\ln x - \mu}{\sigma}\right)^2\right)}{x\sigma\sqrt{2\pi}}\sigma, \mu(\sigma > 0)$ (14)

Log Logistic (LogL) $f(x) = \dfrac{\mu(\frac{x}{\sigma})^{\mu-1}}{\sigma[1 + (\frac{x}{\sigma})^\mu]^2}$ (15)

GEV $f(x)$

$= \begin{cases} \frac{1}{\sigma}\exp(-(1+k(\frac{x-\mu}{\sigma}))^{\wedge-\frac{1}{k}})(1+k(\frac{x-\mu}{\sigma}))^{-1-\frac{1}{k}} & k \neq 0 \\ K, \sigma, \mu(\sigma > 0) \\ \frac{1}{\sigma}\exp\left(-(\frac{x-\mu}{\sigma}) - \exp(-((x-\mu)/\sigma))\right)k = o \end{cases}$, (16)

where $\sigma$ is the scale, $\mu$ is location, and $k$ is the shape parameter.

## 2.6 Uncertainty estimation and decomposition

We used two approaches to uncertainty analysis in future flood hazard. First is the additive chain approach, which is similar to sensitivity analysis, whereby the main goal is to assess components of the modelling chain independently (one at a time). Second, we used ANOVA to separate the variance contribution of each component of the modelling chain and their interaction. Unlike the former (sensitivity approach), it helps us to understand the interaction of factors and their main variables using their total variance values.

### 2.6.1 Additive chain approach

We examine the relative contribution of four key components of the modelling chain to the projected uncertainty in future flood hazard. These are climate models (CM), bias correction techniques (BC), hydrological model parameters (HP) and flood frequency distribution models (FF). We only use a single SSP scenario (here SSP370) in our evaluation of the uncertainty cascade, with the assumption that different studies will be interested in quantifying flood hazard for different emissions outcomes separately.

Overall uncertainty is estimated based on the additive chain method, while decomposition of the contribution of different sources of uncertainty and their interactions is evaluated using ANOVA. The uncertainty in future flood hazard can be defined as the total uncertainty resulting from the components ($s$) of the modelling chain considered – here the four components $s1$ to $s4$ (CM, BC, HP and FF). Initially we define the cumulative uncertainty for each component ($s$) of the modelling chain denoted by $U^{\mathrm{cum}}(X_1 \dots X_s)$ and its respective conditional probability as $U(q_{xs+1\dots x_s})$. The uncertainty in our modelling chain is characterised as the total variance in the flood quantile generated from each of the four components evaluated, with total uncertainty defined as follows:

$$U^{\mathrm{cum}}(x_1, \dots, x_s) = \frac{1}{\pi_{j=s+1}^{nj}} \sum_{x_{s+1}\in x_{s+1}} \dots \sum_{x_s\in x_s} U(q_{x\,s+1\dots x_s}). \quad (17)$$

$U(q_{x\,s+1\dots x_s})$ stands for the respective cumulative probability of each output quantiles. This uncertainty quantification method keeps summing as the number of components ($s$) of the modelling chain increases. Therefore, the uncertainty of a particular step along the modelling chain, e.g. $s = 4$, denoted by $U^{\mathrm{cum}}(X_4)$, can be described as follows:

$$U^{\mathrm{cum}}(X_4) = U^{\mathrm{cum}}(X_4) - U^{\mathrm{cum}}(X_3) \text{ in general}$$
$$U^{\mathrm{cum}}(X_s) = U^{\mathrm{cum}}(X_{s-1}) - U^{\mathrm{cum}}(X_{s-2}). \quad (18)$$

Note that uncertainty of each component of the chain is a magnitude of contribution to the total uncertainty so that the sum of uncertainties of individual sources is always equal to the cumulative uncertainty $U^{\mathrm{cum}}(X_1, \dots, X_s)$, whereas, Eq. (18) was used to obtain the individual component from the total uncertainty.

### 2.6.2 ANOVA approach

Unlike single-chain-based additive approaches to uncertainty estimation, ANOVA can decompose the aggregated source of uncertainty into individual components and their interaction, using specific extreme flow indices (Meresa and Romanowicz, 2017), and in mean runoff projections (Bosshard et al., 2013). We develop a hypothesis test using an ANOVA model that can identify the effect of each component (CM, BC, HP and FF) in the model chain to the total variance of the extreme index ($Y$). According to $n$-way ANOVA principles, the model splits the total sum of squares (SST) into the sum of squares (SS) of the main variables and their interactions as follows:

$$\mathrm{SST} = \sum_{i=1}^{\mathrm{NCM}=12}\sum_{j=1}^{\mathrm{NBC}=5}\sum_{k=1}^{\mathrm{NHP}=300}\sum_{l=1}^{\mathrm{NFF}=3} (Y_{ijkl} - \overline{Y})^2, \quad (19)$$

where $Y_{ijkl}$ is estimated flood magnitude considering $i = 12$ climate models, $j = 5$ bias correction methods, $k = 300$ hydrological parameter sets and $l = 3$ extreme frequency distributions, and $\overline{Y}$ is the mean of all variables. SST is the grand square deviation of the main and interacting variables. Furthermore, the deviation in SST is split into individual and interacting components to explore their effect on the aggregated extreme flood frequency indices as follows:

$$\mathrm{SST} = \mathrm{SS_{CM}} + \mathrm{SS_{BC}} + \mathrm{SS_{HP}} + \mathrm{SS_{FF}} + \mathrm{SS_{CMBC}}$$
$$+ \mathrm{SS_{CMHP}} + \mathrm{SS_{CMFF}} + \mathrm{SS_{BCHP}} + \mathrm{SS_{BCFF}}$$
$$+ \mathrm{SS_{HPFF}}, \quad (20)$$

where $SS_{CM}$ is the sum of standard errors of climate models, $SS_{BC}$ is the sum of standard errors of bias correction methods, $SS_{HP}$ is the sum of standard errors of hydrological parameters, and $SS_{FF}$ is the sum of the standard errors of flood frequency. In examining combined effects, $SS_{CMBC}$ is the sum of standard errors of climate models and bias correction methods, $SS_{CMFF}$ is the sum of standard errors of climate models and extreme frequency, and $SS_{BCHP}$ is the sum of standard errors of bias correction methods and hydrological parameters.

## 3 Results

### 3.1 Evaluation of bias correction techniques

The annual maximum daily precipitation series from 12 CMs were evaluated against observations for each catchment during the period 1976–2005 (reference period). RR, MAE, PBIAS and RMSE results for each catchment are presented in Figs. 3 and S1. Following bias correction, results indicate an improvement in the CMs in reproducing observed annual maximum precipitation. However, the performance of each bias correction method is not uniform for all catchments and CMs. MAE values range from 0 to 195, RMSE from 0 to 40, PBIAS from −65 to 40 and RR ranges from −0.40 to 0.50. Overall, the distribution-based bias correction methods performed better in reproducing observed maximum precipitation in these catchments. The smallest PBIAS is observed in the Boyne catchment and largest in the Newport catchment. The BC-CCSM (Beijing Climate Center Climate System Model) and ACCESS-CM2 (Australian Community Climate and Earth System Simulator coupled model) climate models show high RR (0.4) in the Slaney catchment using DGQM, SGQM and BSM, whereas GFDL (Geophysical Fluid Dynamics Laboratory Coupled Model) and MPI-ESM-LR (Max Planck Institute Earth System Model) climate models indicates lower performance in correcting precipitation using all bias correction methods. Figure 4 shows the raw GCM outputs together with the results of the five bias correction methods applied to the CMIP6 ensemble in simulating the maximum daily precipitation in each month for each catchment. The corrected precipitation gives a wider spread in winter months (except DGQM and EQM techniques), relative to summer months. However, the ensemble spread is dependent on the bias correction method. DGQM and EQM methods result in a relatively narrower spread, whereas SGQM, BSM and CF return a wide range of simulations (Fig. 4).

The influence of each bias correction method on the magnitude of simulated changes in annual maximum daily precipitation for the 2050s (2040–2069) and 2080s (2070–2099; relative to reference period 1976–2005) was evaluated for each GCM SSP combination (Fig. 5). Simulated changes from the same 12 CMs using different bias correction approaches show substantial differences in the magnitude of changes in annual maximum precipitation. Generally, projected changes are smaller in the Slaney and Boyne catchments using DG (double gamma) methods, whereas the changes in the Blackwater and Newport are smaller when using EQ (empirical quantile) and SG (single gamma). There is also a linear relationship between the annual maximum precipitation changes suggested for the 2050s and 2080s for most bias correction techniques and catchments, indicating a positive trend in the magnitude of changes with time (Fig. 5). However, we note that there is greater scatter for EQM relative to other bias correction techniques and in the Blackwater catchment relative to other catchments. We also note that the magnitude of change in annual maximum daily precipitation tends to be higher for SSP585 than for SSP370 and SSP126 and slightly higher in the Slaney and Newport catchments relative to the Boyne and Blackwater catchments. Changes in annual maximum daily precipitation for SSP126 are smaller than those simulated for SSP370 and SSP585 due to the lower radiative forcing and socio-economic impact associated with SSP126 relative to other scenarios.

### 3.2 Hydrological modelling and parameter uncertainty evaluation

The GLUE procedure was used to identify a set of model parameters that offer acceptable performance. In total, 30 000 parameter sets were randomly generated from uniform distributions with the NSE objective function threshold of > 0.5 calculated from daily mean flows used to identify behavioural parameter sets. To produce future simulations, we employ the top 300 behavioural parameter sets. Figure S3 shows the variation in model parameter values and NSE for each GR4J model parameter and catchment. The majority of parameters show clearly identified peaks for each catchment, particularly the parameters $X2$, $X3$ and $X4$. The only parameter to show a lack of identifiability is the $X1$ parameter (the production storage capacity) in particular in the Newport catchment. This is the smallest of the four catchments and displays large variability in topography and land cover (forestry) and has a lake present in the catchment. Figure 6 shows the ability of the GR4J model to simulate the maximum daily flows in each month using behavioural sets relative to observations for the years 1976–2005, together with the 95 % confidence intervals of the simulations. Simulations successfully capture the observations, which mostly fall within the 95 % confidence interval, except in the Boyne catchment, where a portion of the observed maximum flow time series falls outside of the simulated confidence interval. We note that the Boyne catchment has undergone significant arterial drainage works, which is likely to result in higher peak flows (Harrigan et al., 2014) in the observations relative to the model simulated values. GR4J achieved the best fit in the 95 % CI modelling discharge in the Newport catchment. Overall, the model was able to capture winter flows well in

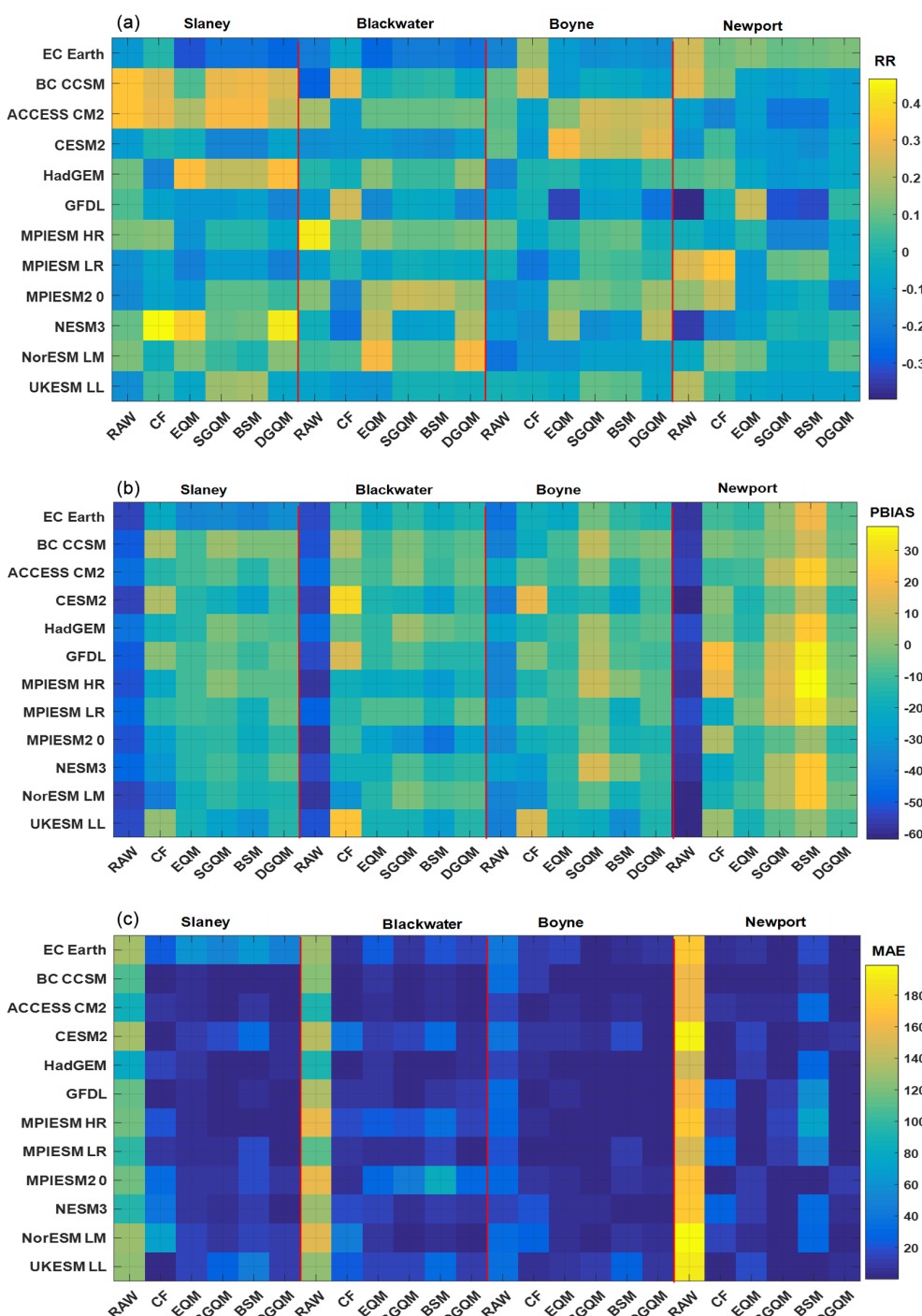

**Figure 3.** Comparison of corrected annual maximum daily precipitation using five bias correction methods and observed annual maximum precipitation using the **(a)** Pearson correlation coefficient (RR), **(b)** percent bias (PBIAS) and **(c)** mean absolute error (MAE). The *y* axis indicates each of 12 CMIP6 climate models, and the *x* axis represents each of the five climate bias correction methods.

all catchments, particularly in winter peak flow simulations. The biggest discrepancies were observed in modelled flows during late summer, with poorer performance likely associated with use of the NSE objective function which is more sensitive to the large peak values than lower flows in summer.

We examined the uncertainty associated with model parameters for different CMs and future time periods. Figure 7 shows the simulated annual maximum daily flow for the

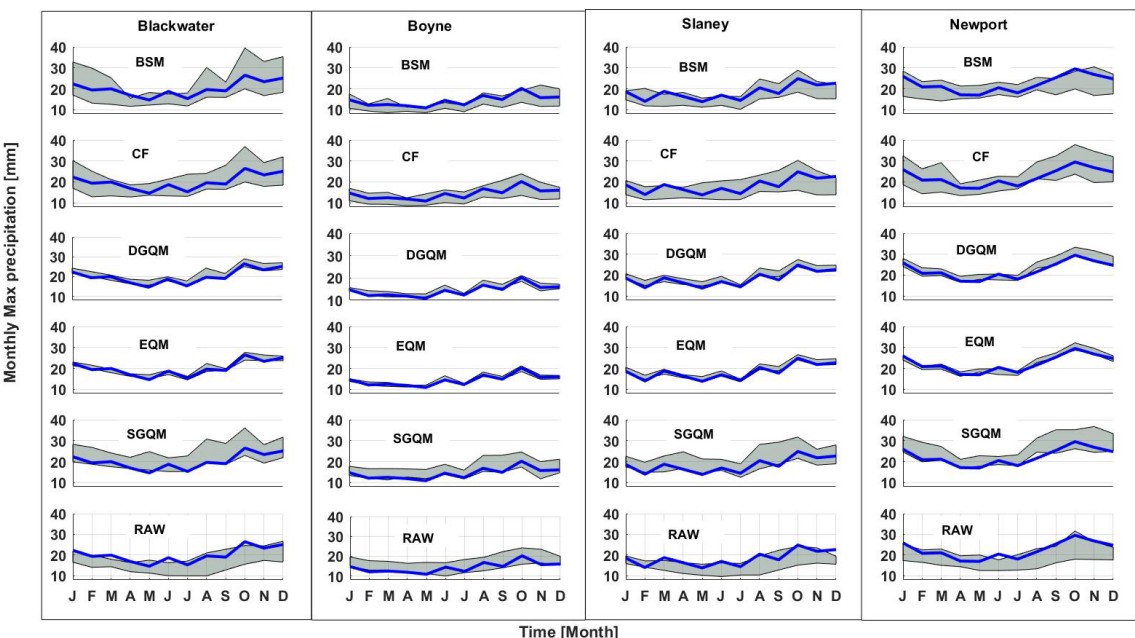

**Figure 4.** Comparison of raw and bias-corrected simulations from 12 CMIP6 GCMs with observed maximum daily precipitation in each month for each of our four study catchments. Each row presents results of one of five bias correction techniques (top row – Birnbaum distribution (BSM); second row – change factors (CF); third row – double quantile mapping using gamma distribution (DGQM); fourth row – empirical quantile mapping using simple interpolation (EQM); fifth row – single quantile mapping using gamma distribution (SGQM); last row – raw GCMs output). In each panel the grey shaded region represents the spread of 12 GCM simulations, with blue line being the observed maximum precipitation.

reference and each future time period using the 300 GR4J behavioural parameter sets forced with 12 CMs bias corrected using DGQM. It is evident that the uncertainty associated with hydrological model parameters is not static and varies depending on climate model, catchment and future time period. The importance of hydrological model uncertainty tends to be greatest for climate models that project the largest magnitude of change, indicating that the changing characteristics of precipitation (e.g. mean and variability) have a large bearing on the importance of hydrological model uncertainty. It is also likely that different bias correction techniques interact with parameter uncertainty. The uncertainties associated with hydrological model parameters tend to be greatest for the 2080s (black lines in Fig. 7) and for the Boyne and Slaney catchments.

Percent changes in maximum annual daily flow are simulated using behavioural parameter sets and 12 CMs bias corrected by each of the five methods, together with the raw simulations are presented in Fig. 8. A clear increase in the annual maximum flow is suggested using DGQM, SGQM, CF, EQM and BSM; however, the magnitude of simulated increases differs depending on bias correction approach and is not uniform across the selected catchments. Overall, distribution-based bias correction models give smaller changes in annual maximum flow time series. This indicates that wet day frequency correction is very important for understanding future

annual maximum flow projections. Annual maximum flow changes using EQM gives a higher spread range and uncertainty in each catchment, while the change factor method produces the most modest changes. Changes in annual maximum flow are smallest in the Blackwater, which is notable as the catchment with largest groundwater storage. Changes in annual maximum flow are more constrained under SSP126 than SSP370 and SSP585; however, differences between SSPs are modest, and uncertainty ranges of changes in annual maximum flow are considerable for each one (Fig. 8).

## 3.3 Flood quantile projections under varying climate conditions

Figure 9 shows the estimated flood quantile values at different return periods (from 2 to 100 years), using GEV, LN and log-logistic (LogL) distribution models, and associated 95 % confidence intervals from 12 CMs (under the SSP370 scenario) bias corrected using the DGQM method over the full simulation period (1976–2100). Overall, the LN distribution returned the smallest flood quantile magnitude, while the estimated quantile values using LogL tend to be largest across each catchment and have the narrowest uncertainty bands. The GEV distribution produces the largest uncertainty ranges in flood quantiles in all catchments (Fig. 9).

Percent changes in flood quantiles for return periods ranging from 10 to 50 years were also evaluated for each future

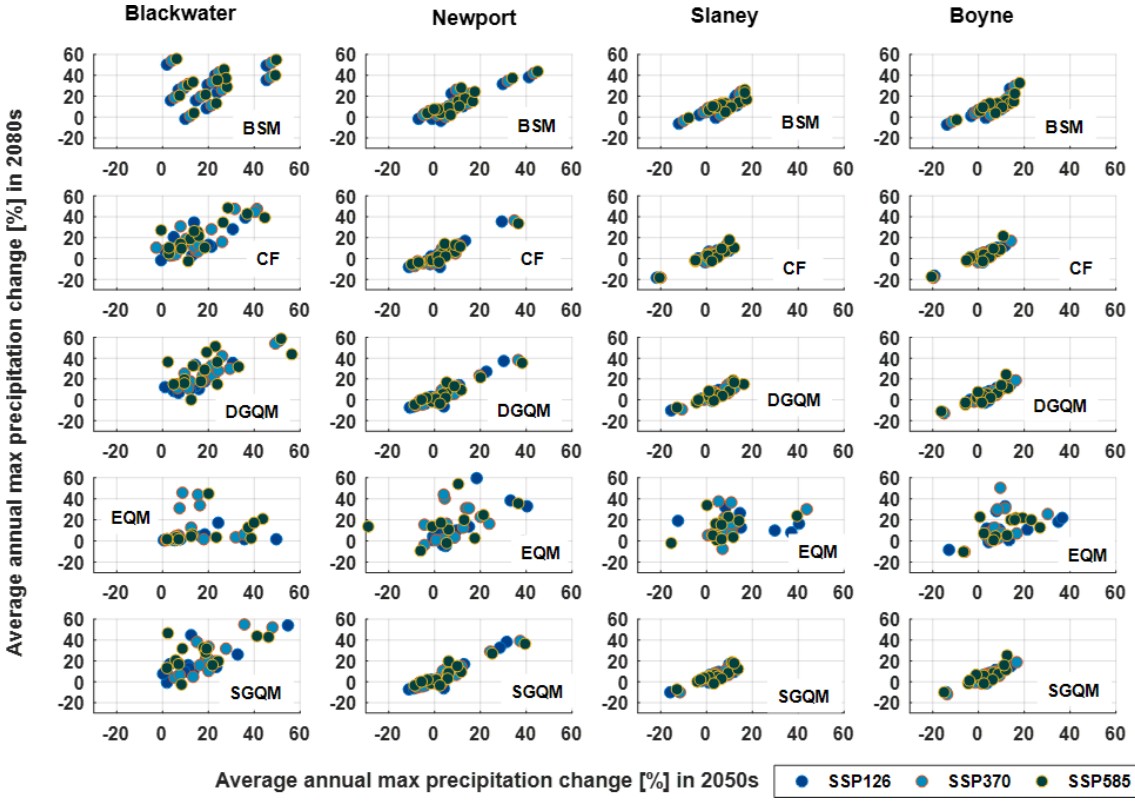

**Figure 5.** Scatterplots of the change in maximum annual daily precipitation as simulated by 12 CMIP6 GCMs (each dot) and corrected using each of the five bias correction techniques for three SSP scenarios for the 2050s (2040–2069) and 2080s (2070–2099). The dark blue circle represents SSP126, light blue represents SSP370, and dark green represents SSP585.

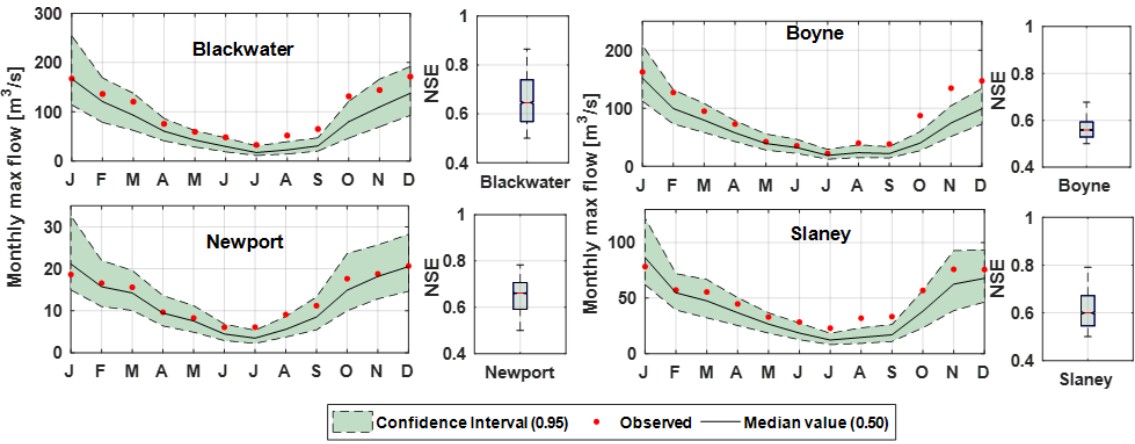

**Figure 6.** Observed and simulated monthly maximum daily flows (cubic metres per second; hereafter m$^3$/s) using the GR4J behavioural parameter sets for the reference period (1976–2005). The black line and shaded area represent the median and 95 % confidence interval of simulated flows, respectively, and the red dot represents observations. Box plots show the spread in NSE scores for behavioural parameters sets in each catchment.

period under SSP370, using an ensemble mean of three distribution types fitted to the median GR4J simulation combining all CMs and bias correction methods (Fig. 10), and Figure S4 shows the quantile values using the GEV distribution fitted to the median GR4J simulation combining all

CMs and DGQM bias correction methods (Fig. S4). Each catchment shows a significant increase in the magnitude of flood quantiles in the future. Overall, changes in flood quantiles are consistent with changes in annual maxima daily flow and changes depending on the climate period (2020s, 2050s

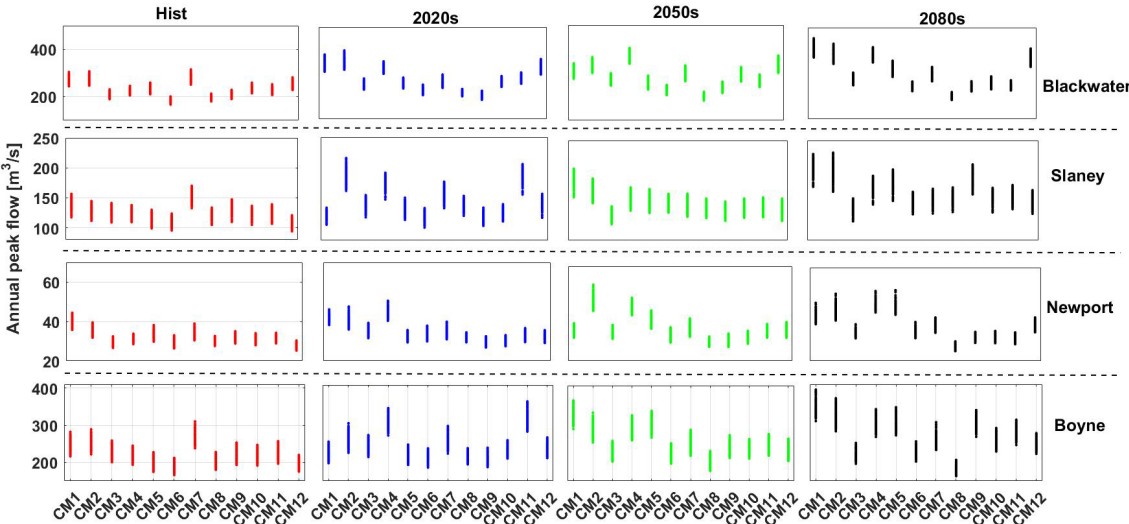

**Figure 7.** Hydrological model parameter uncertainty in simulating the annual maximum daily flow in each catchment using the best 300 parameter sets for each climate model for the reference period (red), the 2020s (green), 2050s (blue) and 2080s (black) under the SSP370 scenario and DGQM bias correction method.

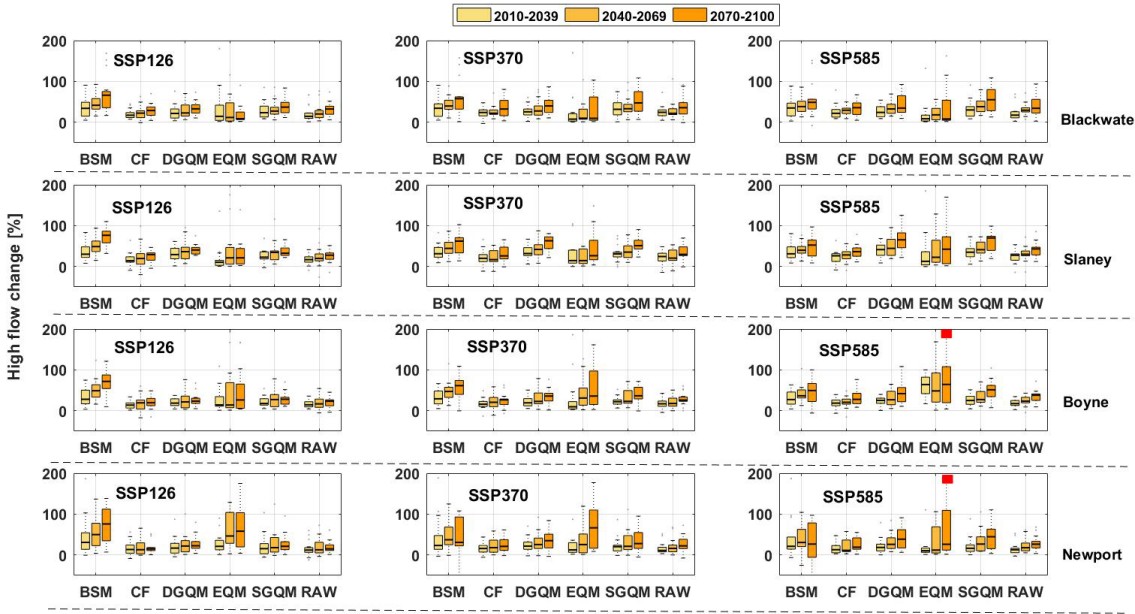

**Figure 8.** Matrix of projected percent changes in annual maximum daily flow for the 2020s (2010–2039), 2050s (2040–2069) and 2080s (2070–2099) for SSP126, SSP370 and SSP585 in each catchment with respect to the reference period (1976–2005). Each box plot represents the spread of 12 GCMs for each of the five bias correction techniques, as simulated using behavioural model parameter sets. Marked for each box plot are the median and 0.25 and 0.75 quantiles. The red dots are outliers (changes above 200 %).

and 2080s). However, future flood changes are not the same across catchments, and differences exist based on the bias correction methods employed. The largest changes in flood quantiles across all catchments are found using the BSM bias correction method, where, in the Blackwater catchment, increases of over 90 % are found for the 50-year return period in the 2050s and 2080s. Other bias correction techniques re-

turn more modest increases. Changes in flood quantiles estimated using the CF and DGQM approaches show larger changes in the 2050s relative to the 2080s in the Blackwater. Indeed, the CF method shows the most modest changes in all quantiles for the 2080s relative to other time periods across all catchments. In the Slaney catchment, most bias correction methods show only small changes in flood quantiles between

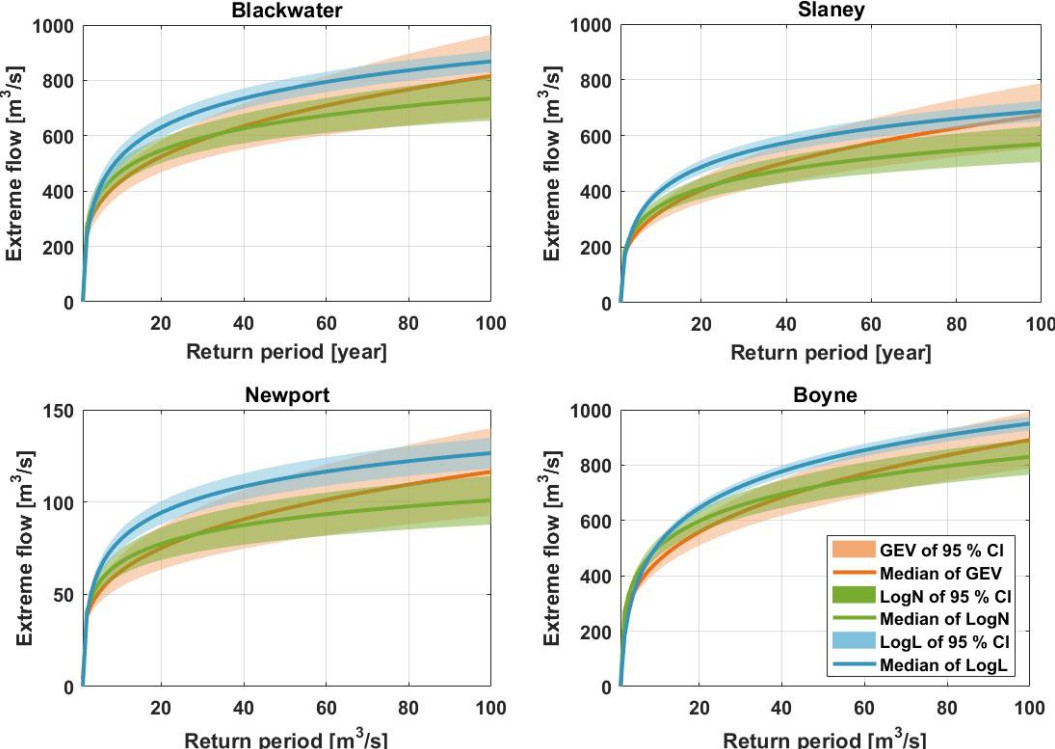

**Figure 9.** Comparison of flood quantiles estimated using three flood distribution models (log-logistic, GEV and LN) fitted to projected annual maxima series derived from 12 GCMs bias corrected using the DGQM method. The shaded region indicates the 95 % confidence interval of the annual maximum projections from 12 GCMs under the SSP370 scenario over the full projection period (1970–2100 period) in each catchment.

time periods, with the exception being EQM, which shows substantial increases in each flood quantile moving across each time period. The EQM method tends to show the greatest increases in magnitude for higher quantile floods, with the largest percent changes returned for the 50-year return period. In the Newport and Boyne catchments, the CF approach shows modest reductions in the magnitude of higher-frequency floods, a characteristic not reported using other bias correction techniques. Interestingly, the changes depend on the future climate period, with most catchments showing higher change in the 2020s than the 2050s. This is likely due to climate variability and nonlinearity in the climate response.

### 3.4 Uncertainty estimation and decomposition for flood quantile estimation

The magnitude of each of the four sources of uncertainty on flood hazard estimation was estimated first based on the additive chain principle to reveal their contribution to the overall uncertainty in projected changes. Projections from each of the 12 climate models were iteratively passed through five bias correction techniques and used to force the GR4J model using behavioural parameters sets before fitting each of the three frequency distributions to estimate future flood

hazard. Therefore, the sensitivity to climate model uncertainty was evaluated by using each of the 12 CMs together with a single bias correction technique, median hydrological model parameter set and a single flood frequency distribution. Figure 11 presents the integrative range of this cascade of uncertainty in estimating return period floods for each catchment using data spanning the years 1976–2100. The dominant source of uncertainty differs from catchment to catchment. Climate models, flood frequency distribution and bias correction techniques tend to be the dominant contributing source of uncertainty in estimating flood quantiles. Using the one-at-a-time approach to evaluating uncertainties in flood quantiles, climate model projections in the Newport and Blackwater catchments present the dominant source of uncertainty, particularly for higher flood quantiles. Notably, the relative magnitude of uncertainty in the smaller flood quantiles in the Blackwater catchment is different than for larger flood quantiles, with the extreme value distribution becoming dominant at lower flood quantiles. The largest source of uncertainty in the Slaney is derived from the use of different frequency distribution models. For the Boyne catchment, the bias correction methods contribute more to the total uncertainty. Therefore, the various steps in the modelling chain contribute differently in each catchment to total uncertainty

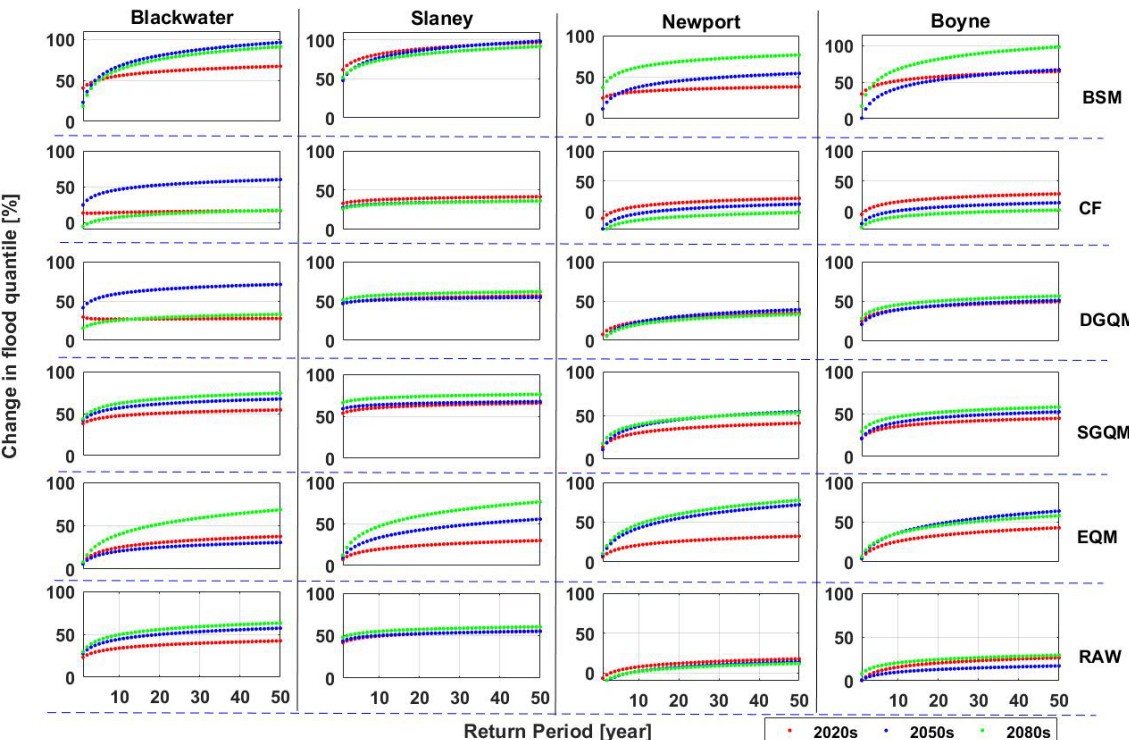

**Figure 10.** Percent changes in extreme flow quantiles using the ensemble mean of three distribution types, 12 GCMs and 300 parameter sets in each catchment for the 2020s (red), 2050s (blue) and 2080s (green). Simulated changes are derived using raw (bottom row) and bias-corrected (the first four rows) simulations.

in future flood hazard estimation and, in some catchments, vary, depending on the flood quantile of interest (Fig. 11).

While additive chains may be useful for evaluating individual components of uncertainty, they do not allow identification of the interactive components of the uncertainty cascade. We use ANOVA to decompose the contribution of individual sources of uncertainty and their interaction in contributing to the total uncertainty in future flood hazard (Fig. 12). We evaluate changes in the 10-, 50- and 100-year return period flood for the 2080s (2070–2099) relative to the reference period (1976–2005). The decomposition of uncertainty is based on the variance in changes of flood quantile values and calculated out of 100 %, i.e. the sum of all sources is equal to 100. The contribution of each source of uncertainty is not uniform across the selected catchments (Fig. 12). Taking the example of the magnitude of the 100-year flood (Fig. 12; outer band), future flood hazard in the Blackwater catchment for the 2080s is highly sensitive to the climate model used (36.5 % of total uncertainty in 100-year flood) and respective bias correction methods (20 % of total uncertainty). In contrast, future flood frequency in the Boyne catchment is more sensitive to the bias correction methods (30.0 %) and extreme value distribution (25.5 %) employed. In the Boyne catchment, the climate model ensemble accounts for less than one-quarter (22.5 %) of the uncertainty in simulations of the 100-year flood by the 2080s and is compa-

rable in magnitude to the interactive components of the cascade. In the Newport catchment, the use of different extreme value distributions (36.3 %) presents the largest component of uncertainty, followed by the climate model ensemble. The results of the ANOVA also confirm that the uncertainty resultant from the interaction of the key sources considered is not negligible and also varies on a catchment basis. In the Slaney and Blackwater catchments, the interaction of various uncertainty components accounts for 16.8 % and 16 % of the total uncertainty in estimates of the 100-year flood by the 2080s. In contrast, interactive components in the Newport and Boyne catchments account for approximately one-quarter of the uncertainty range (25.2 % in the Newport and 22 % in the Boyne catchments). Overall, hydrological parameter uncertainty is the least dominant source of uncertainty in future flood frequency estimates across each catchment (Figs. 11 and 12). For estimates of the 100-year flood in the 2080s (Fig. 12; outer band), hydrological model parameter uncertainty is typically 5 % of the total uncertainty, reaching up to 9 % in the Boyne catchment. Furthermore, the contribution of each of the main components of uncertainty and their interactions to total uncertainty is broadly similar for the different return periods considered for our analysis of the 2080s (Fig. 12).

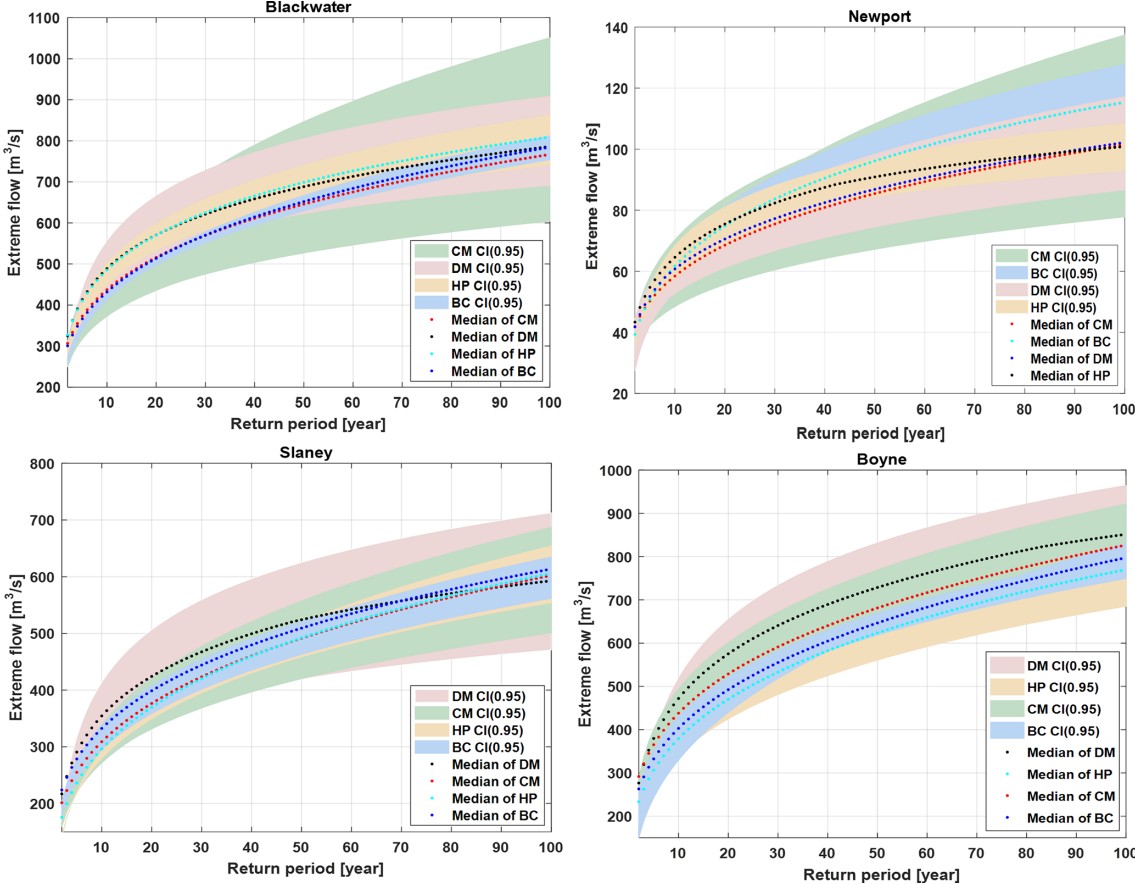

**Figure 11.** Total uncertainty in flood frequency curve using 12 CMIP6 GCMs (CM), five bias correction methods (BC), 300 behavioural parameter sets (HP) and three extreme value distribution models (DMs). Shaded areas represent the 95 % confidence interval for each component in the modelling chain, while the dotted line stands for their respective median values. The curves are fitted to annual maximum time series derived for 1976–2100.

## 4 Discussion

We examined future changes in flooding for four catchments in Ireland, including within our modelling chain the uncertainties derived from 12 CMIP6 CMs, five commonly used bias correction techniques, hydrological model parameters and three flood frequency distributions. We found that maximum precipitation in all catchments is projected to increase over the coming decades; however, changes are not uniform across catchments. The largest change in annual maximum precipitation was found for the Newport catchment, with ensemble median increases becoming progressively larger for the 2020s, 2050s and 2080s (27 %, 40 % and 60 %, respectively). In the Slaney catchment, increases in annual maximum precipitation between the 2050s and 2080s are not statistically significant. In the Blackwater catchment, most of the climate models indicate a positive change, but these have a wide range of uncertainty relative to other catchments (from 0 % to 55 % in 2050s and 0 % to 60 % in 2080s). We find projected changes in flood magnitudes to be broadly proportional to changes in maximum precipitation in all catch-

ments. Projected changes in flood magnitude are higher using the SSP585 and lower using SSP126 climate scenarios. Overall changes in precipitation and air temperature are broadly consistent with previous findings using CMIP5 (Broderick et al., 2019) in Ireland.

Our findings demonstrate the large uncertainties associated with projected flood magnitudes. Over recent years, much research has sought to explore the dominant sources of uncertainty in climate change impact assessments. Similar studies have identified climate models as being the dominant source (e.g. Sulis et al., 2012; Hattermann et al., 2018). Indeed, Addor et al. (2014) highlight that there seems to be a general agreement in the literature on the dominant contribution of climate models to the uncertainty in discharge projections. In contrast, our findings demonstrate the challenges of generalising about the dominant components of the cascade of uncertainty at the catchment scale, important for understanding future flood risk. Using both additive chains and variance decomposition to examine components of the modelling chain employed, our results show that not only does the

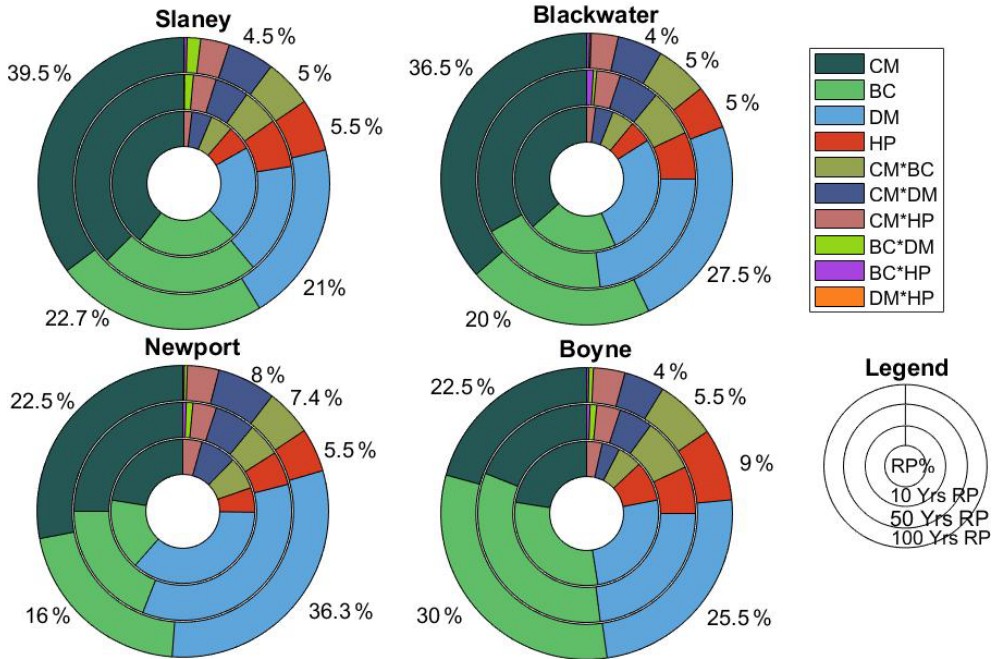

**Figure 12.** Contribution of each component of the modelling chain, together with their interaction, to the range of simulated changes in flood hazard magnitude for the 2080s (2070–2099) relative to the reference period (1976–2005). Sources considered include 12 CMIP6 climate models (CM), five bias correction (BC) techniques, hydrological model parameter uncertainty (HP) and three extreme value distribution models (DMs). The outermost circle/doughnut represents the flood quantile change at return period (RP) of 100 years, the middle circle RP of 50 years and innermost circle RP of 10 years for the 2080s. Percent changes are annotated for the outermost circle (100-year flood).

contribution and importance of different components of the modelling chain vary across catchments, but that the dominant source of uncertainty can be completely different. Moreover, the dominant sources of uncertainty vary depending on which approach is used to evaluate uncertainty. Using additive chains, in two of our study catchments the climate model ensemble was the dominant source of uncertainty, as was expected. However, for the Boyne and the Slaney catchments, the extreme value distribution employed for flood frequency analysis and the bias correction method used to adjust raw GCM output contribute greater uncertainty to future flood risk than the ensemble of CMs employed. Using variance decomposition allows for the interaction of components of the modelling chain to be evaluated and quantified and offers a fuller assessment of uncertainty. Our results show that in the Slaney and Blackwater catchments the climate model ensemble used dominated the cascade of uncertainty. In the Slaney catchment, the second most important component is the bias correction technique used, while in the Blackwater catchment the frequency distribution model is the second most important component. In the Newport catchment, the flood frequency distribution model ranks as the most important component of the cascade of uncertainty, followed by the climate model ensemble. Finally, in the Boyne catchment, the climate model ensemble ranks as only the third most important component of the uncertainty cascade in simulating

flood quantiles, with the bias correction methods coming first and the flood frequency distribution second.

We hypothesise that catchment characteristics may influence the dominant components of the cascade of uncertainty in each catchment. The Slaney and Blackwater catchments, where climate models dominate, are among the largest in our sample, with groundwater contributing significantly to runoff in both. In the Blackwater catchment, topography has a large bearing on flood response time within the catchment, with flood peaks moving more rapidly from elevated subcatchments into the main channel, and may be a reason as to why the flood frequency model ranks as the second dominant component in the cascade of uncertainty. The Slaney catchment drains the southern component of the Wicklow Mountains before reaching shallower elevations as it flows south through Carlow and Wexford. The variation in topography and, hence, precipitation from upper to lower reaches may be a reason as to why bias correction methods emerge as the second most important component of the uncertainty cascade in this catchment. The Newport catchment is the smallest in our sample, with the flood response influenced more strongly by varied land use cover (forestry and rotational cropping) and the presence of a lake in the upper reaches of the catchment. The flood frequency distribution dominates the cascade, followed by the climate model ensemble used. Finally, the Boyne catchment has been heavily influenced by arte-

rial drainage works, resulting in the insertion of field drains and the deepening and widening of the river channel (Harrigan et al., 2014). These interventions have resulted in a faster rainfall–runoff response. In the Boyne catchment, therefore, projected changes in future rainfall are likely to have a more direct bearing on the flood response, with uncertainty in bias correction and the flood frequency distribution dominating the cascade of uncertainty.

Across each of our catchments and in the context of the modelling chain employed, the uncertainty from hydrological model parameters is the smallest component of total uncertainty in the estimation of future flood hazard magnitude. Bastola et al. (2011a, b) identify hydrological model uncertainty as being important in a previous assessment of change in flood risk, while Mizukami et al. (2016) rank hydrological model uncertainty to be of greater importance than bias correction approaches in USA catchments. We note that we do not consider hydrological model structure uncertainty in our modelling chain, and that our decision to select only the top 300 behavioural parameter sets, rather than all those identified, may be a factor here. This highlights how subjective decisions in the application of hydrological models may have important implications for uncertainty in future projections (Melsen et al., 2019). We also find that the contribution to total uncertainty from the interaction of components within the modelling chain is substantial – up to one-quarter of the uncertainty range considered in two catchments.

While uncertainty decomposition may be useful in helping scientists navigate the cascade of uncertainty by prioritising the key components of the uncertainty cascade to sample (Clark et al., 2016) for a particular catchment, our small sample makes it difficult to determine how catchment characteristics may be informative in identifying the dominant sources of uncertainty a priori. There are some indications that catchment properties and the characteristics of observed precipitation may influence which uncertainties may dominate. Vetter et al. (2017) used variance decomposition to examine the contribution of modelling chains (GCM, emissions scenario – in the form of Representative Concentration Pathways or RCPs – and hydrological model) to total uncertainty in different parts of the flow regime. In their study, they examine 12 large catchments globally. While Vetter et al. (2017) highlight the dominance of CMs in total uncertainty across many catchments, they also highlight that the share of emissions scenario and hydrological model contributions to overall uncertainty can differ between catchments. Similar to our findings, Vetter et al. (2017) highlight the importance of interacting components of the modelling chain employed.

As our study has shown, the evaluation of uncertainties and their interaction in future flood risk assessment with climate change is a critical step in understanding future impacts and informing adaptation decision-making. However, results depend on the modelling chain employed and the method used to quantify uncertainties. Given the possibility of examining interactive components of a modelling chain,

we recommend the use of ANOVA-based techniques to decompose the cascade of uncertainty. We highlight, however, that our modelling chain is not complete, and that the work presented here is illustrative rather than exhaustive. We do not, for example, include uncertainties in future flood estimates derived from different hydrological model structures with previous work, as indicated above, showing that hydrological model structure and complexity may be an important component of future uncertainties. Other potential sources of hydrological model uncertainty are also omitted. For example, we neither evaluated the transferability of parameter sets to represent the future rainfall–runoff response (e.g. Broderick et al., 2016) nor did we evaluate the influence that selection of different objective functions may have on the identification of behavioural parameter sets. Westerberg et al. (2020) also highlight the potential importance of uncertainty in the discharge data used to calibrate hydrological models. In assessing different flood quantiles, we use only 30 years of annual maxima to extrapolate to return periods outside our record length. Such extrapolations should be treated cautiously but are prevalent in many studies (e.g. Kay et al., 2006; Lawrence, 2020; Meresa and Romanowicz, 2017), while rarer floods are also associated with design standards (e.g. 100-year flood) and are therefore of interest to decision-makers. We also assume that flood processes remain stationary within the 30-year windows used to evaluate changes, while other factors that can influence rainfall–runoff response, such as land use change, remain unchanged from the calibration period used to train our hydrological models.

## 5 Conclusions

This study evaluates changes in future flood magnitude with climate change for four Irish catchments using a modelling chain incorporating 12 CMs comprising the CMIP6 ensemble, five bias correction techniques, hydrological model parameter uncertainty and the use of extreme value distributions. Our findings suggest increasing flood hazards over the coming decades in all catchments, with changes in flooding largely consistent with changes in maximum precipitation. However, uncertainties in future flood changes are large and increase with time and flood quantile. Using ANOVA, we decompose uncertainties in future flood quantiles to examine how individual components of our modelling chain and their interactions contribute to overall uncertainty. Our results show how the dominant sources of uncertainty may vary on a catchment basis, highlighting the inability to generalise on the dominant components of uncertainty across catchments. Across all four catchments, the climate models, bias correction methods and the extreme value distributions used to evaluate flood return periods were differentially dominant, while the uncertainty derived from the interaction of various components was substantial in all catch-

ments. Our work shows the value of ANOVA methods in visualising and quantifying the uncertainty cascade at a catchment level, which will be helpful in navigating the uncertainties associated with future flood risk for better adaptation decision-making. While our sample is small, the dominant components of uncertainty in future flood risk may be related to catchment characteristics. We, therefore, recommend that future work seeks to better understand the link between the key components of the cascade of uncertainty and catchment characteristics. The ability to do so would offer considerable advantages in knowing a priori which sources of uncertainty should be targeted as priority.

*Code availability.* The codes used and developed during the current study are available from the corresponding author on reasonable request.

*Data availability.* The data sets generated during and/or analysed during the current study are available from the corresponding author on reasonable request.

*Supplement.* The supplement related to this article is available online at: https://doi.org/10.5194/hess-25-1-2021-supplement.

*Author contributions.* HM designed the study with input from CM and RF. HM conceived the model and the computational framework and analysed the results. HM wrote the paper with input from all authors.

*Competing interests.* The authors declare that they have no conflict of interest.

*Acknowledgements.* We thank the data providers and three reviewers for their constructive comments and feedback which helped improve our paper. This work was made possible by the award of funding from the Office of Public Works and Kildare County Council to Hadush Meresa, Conor Murphy and Rowan Fealy.

*Financial support.* This research has been supported by the Office of Public Works and Kildare County Council.

*Review statement.* This paper was edited by Nadav Peleg and reviewed by Francesca Pianosi, Martina Kauzlaric, and one anonymous referee.

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
