# Peer review of "Uncertainties and their interaction in flood hazard assessment with climate change"

_Hydrology and Earth System Sciences, 2020_

## Referee Comment (RC1) · Anonymous Referee #1 · 26 Jan 2021

General comments:

The manuscript investigates the uncertainty contributions from different components of the modeling chain in future flood magnitude of four Irish catchments. Using ANOVA, the study considers several sources of uncertainty: GCMs, bias-correction, hydrological model parameter, and extreme value distribution and their interactions. Although the manuscript is generally well written and figures are clear, I think it ends up being merely descriptive and is somewhat incomplete as it does not tackle the reasons why uncertainty varies across catchments. In fact, the main finding I retained after reading the paper was that hydrological model parameter is the least important source of uncertainty (I also don't think there is novelty in stating that sources of uncertainty vary across catchments as emphasized in the conclusions). As the authors noted in the

manuscript, the number of catchments analyzed is small, therefore results do not allow to pinpoint aspects that can further our understanding of why uncertainty shares vary across catchments. The authors themselves write in the final phrase of the conclusions something that I think they should have tackled in their work: "Future work to better understand the link between the key components of the cascade of uncertainty and catchment characteristics is therefore recommended".

In my opinion, this work could improve through the following:

- The authors could take the advantage of having scrutinized three variables (Temperature, Precipitation, and runoff) in the control period, using observed data, to select model runs that behaved better, and use them to estimate future magnitudes. This would provide valuable information, because though limited to few catchments it would show the value of forming constrained chains of models and to assess how these compare to the original "big ensemble of runs" in terms of the uncertainty that is associated to return periods of future floods. For instance, once the best bias correction methods are identified (e.g. Figure 4, or L. 431 DGQM, SGQM), these could be used to form a constrained ensemble. This could affect results in uncertainty shares and, importantly, wold help reducing the uncertainty.

- Avoid the word risk (In the title too). What is done in this paper does not deal with flood risk, but rather with changes in future flood magnitude.

- A word of caution on the return periods of 100 years, based on periods of 30 years. Extrapolation of extreme value fits to a domain well outside the range of the reference period (30-yrs) may itself add considerable uncertainty to the estimate. Extreme values are already hard to sample for their rare nature. I would limit return periods to 30yrs to avoid calculation of return values for return periods exceeding the record length. This aspect should be at least discussed.

- Scenarios. Multiple SSP scenarios are employed at one stage, but they are not considered among the different shares of uncertainty (e.g. Fig 13). The authors should

justify this choice.

- Use of variables: variables are used both at daily and at monthly scales. When dealing with events that have short onset and duration like floods. Possibly, I suggest using the daily time scale at all times.

Please find notes on text and figures below.

Technical corrections: L. 30 - Would avoid word "significant" (strong statistical connotation). L. 34 - are shifting -> have shifted L. 53-57 - I would add Shepherd et al. 2018. L. 69-71 - It is stated that "an important step [. . .] contribution of different components of uncertainty and their interaction to be quantified and partitioned to help scientists and decision makers better navigate the cascade of uncertainty". The authors could hint or provide examples on how this information could be used to "better navigate the cascade of uncertainty" and why it is helpful/important. L. 115-130 - Could this go into the introduction? L. 225 - From this sentence it seems like for a given catchment a single distribution is not sufficient to sample extreme events. I think this is misleading as the practice is to test several and then choose the one that fits best. In this case you employ multiple distribution types to assess the uncertainty that is brought about by this source of uncertainty. L. 230-232 - I would swap order in sentence to respect that of the equations that are listed below. L. 239-240 - The assessment of future high flows using multiple scenarios has been done, the authors might add Giuntoli et al. 2018. L. 266 - Why not daily? L. 275-280 - Possible reasons behind the variety of projected changes? L. 285 - Please add "in the range of" after "to be". L. 290 - Please justify the choice of 30000 as number of parameter sets. L. 338 - Reference to Figure 12: I think showing the median does not add much, it is also hard to see it on the figure. L. 375 - "We examined future flood risk". I would rephrase with something like "we examined changes in future flood magnitude". (In line with what is written at line 459). L. 394-395 - I would attenuate the sensational tone of the sentence. L. 397 - I counted one. L 404 - It this stated "results suggest that rather than being the same across catchments" - I don't think anyone has ever presumed that sources of uncertainty remain constant

across catchments, and there are studies in the literature showing this already. Please reformulate. L. 464-465 - rather than "that the dominant sources of unc. vary on a cathment basis" I suggest writing "our results show how dominant sources of unc. may vary on a catchment basis". I don't find the end of the sentence convincing provided that two adjacent catchments can have entirely different hydrological processes that contribute to flood generation - and in turn different shares of uncertainty sources.

References: Shepherd, T.G., Boyd, E., Calel, R.A. et al. Storylines: an alternative approach to representing uncertainty in physical aspects of climate change. Climatic Change 151, 555–571 (2018). https://doi.org/10.1007/s10584-018-2317-9 Giuntoli, I., Villarini, G., Prudhomme, C. et al. Uncertainties in projected runoff over the conterminous United States. Climatic Change 150, 149–162 (2018). https://doi.org/10.1007/s10584-018-2280-5

---

## Referee Comment (RC2) · Anonymous Referee #2 · 4 Feb 2021

The paper analyses the propagation of uncertainty in the estimation of future flood quantiles associated with climate change in four Irish catchments. It uses the ANOVA method to quantify the relative importance of different sources of uncertainties, including the choice of the Global Climate Model, the bias correction approach, the parameterisation of the hydrological model to transform climate variables (rainfall and temperature) into flows, and the choice fo the extreme value distribution for the flood frequency analysis. The analysis is solid and the paper is clearly written and enjoyable to read. The main finding is that the dominant sources of uncertainty vary between catchments. This is not a completely new result, as previous studies already showed that different uncertainty sources are dominant in different places (as also noted in the discussion, L. 390-395). In the discussion, the authors attempt at linking some of these

differences to catchment characteristics and dominant rainfall-runoff processes in each catchment. However, a comprehensive quantification and evaluation of these links is left to future studies. This is a bit regrettable, as I think the identification of these links is not only the logical next step but also the most challanging and potentially most interesting for future climate change impact assessments - if some of those links could be identified, then one could give a priori indications on which uncertainties should be targeted/reduced first, depending on the characteristics of the catchment under study. This said, I still think the manuscript offers an interesting contribution and is worth publishing after some revisions. Below are some suggestions for improvement.

1. The title may be a bit more specific. "Flood risk" may be interpreted as relating to the product of hazard, exposure and vulnerability, whereas this paper only deals with the hazard component. Pereaphs referring to "flood frequency analysis" or "extreme flow magnitudes" in the title would help readers get a better idea of the actual paper content.

2. I think the paper could be made more coincise, in particular the number of figures reduced. I am not sure that all of them are needed to support the key points the authors want to make. Below are some specific suggestions of paragraphs and figures that I think could be removed from the paper and possibly given in Supplementary Material.

3. The Introduction could elaborate a bit more about the ultimate goals of the analysis and previous studies that had similar goals. On Line 69, the authors say "an important step ... is the development of techniques that allow the contribution...". I find this a bit confusing. First, it suggests to me that this paper will deal with "developing" some new technique for sensitivity anaysis, whereas the point here is not to develop techniques but rather use existing ones for a specific application - identify the dominant uncertainties of flood frequency analysis under climate change. Second, it would be good to discuss here what is to be learnt from such sensitivity analysis, how SA results may inform future climate change impacts assessment. These implications were mentioned in the Abstract, I would have expected to read more on this here too, including

the results of previous studies that looked at a similar problem in different catchments (some of these studies are cited in the Discussion on L. 390-395) or studies that used SA to identify dominant uncertainty sources of other hydrological variables. Indeed, that such sources may differ from catchment to catchment has been shown already although pereaphs in different context / for different purposes (see for instance about the variations in dominant parameters of a hydrological model across US cathcments: van Werkhoven et al 2007, Characterization of watershed model behavior across a hydroclimatic gradient, WRR, doi:10.1029/2007WR006271).

4. I do not understand what the "additive and multiplicative approaches to uncertainty estimation" are and how they work. These are mentioned for the first time on L. 240 ("unlike additive or multiplicative approaches to uncertainty estimation, ANOVA....") but no reference is given. I am familiar with SA literature (for example I know ANOVA) and I would not say it is self-evident what the authors are referring to! The results of this "additive" approach are reported in Fig 12. I am not clear how these resultes (the shaded areas) were obtained (Did you let each uncertainty source vary while keeping all others fixed to their 'reference' value? But if so, how did you choose the reference values? Or maybe you progressively added uncertainty sources one at the time? But if so, the order of adding uncertainty probably conditions your results?). Either way: is this analysis really needed? From a methodological point of view, it seems less robust then the following ANOVA and from a practical point of view it lead to the same conclusions (unless I am missing something?) that the relative importance of the uncertain inputs is catchment dependent. So, does it need reporting?

5. Similarly to previous point, I found the explanation of how PCI works and what it is used for (L.210-220) a bit too concise and hence unclear. On the other hand, I am not really sure mentioning it is really needed, as it does not seem to be used in the Results section 3.2. Please clarify or remove.

6. Other figures that the authors may consider moving to Supplementary Materials: Figure 6: is it needed? what for? Indeed text to comment it on L. 282-287 is not

particularly informative. Figure 8: again, what's the point? It shows the ensemble median, so it hardly tell us about the spread of uncertainty, and it brings together past and future behaviour, so it doesnt tell us about the effects of climate change... So, what is the reader expected to learn from it?

7. The authors discuss this already, but I think it is really worth emphasising even more: not considering structural uncertainty in the hydrological model is a major limitation of this study. Results show that parameter uncertainty has small impact on the final output variability, but really we do not know whether that is because the chosen model structure already condition the range of flow predictions that one can get, and whether maybe those would significantly change with a different model structure. This may not be too problematic if the uncertainty about the model structure was small - in other words, if the authors had a strong justification of why that model structure is particularly adequate for the catchments under study. This could be further elaborated in the revised manuscript.

OTHER SPECIFIC POINTS

Squaring in Eq. (10): is it needed? (I would think not)

Sec. 3.1: I got a bit lost and was not completely sure what the key point is here. It seems to me that, regardless of all details, the key point here is that it is not possible to declare one bias correction method to be the best at improving performances in all catchments, so it remains unclear how to choose one; but the choice matters as the projections for the future change quite a bit depending on the method employed.

L. 333: is it sensible that changes are greater in the 2050s than in the 2080s? Isn't it odd?

Figure 13: maybe could be made easier to read by inserting the return period in the circle and explaining acronyms in the legend

606, 2020.

---

## Referee Comment (RC3) · Martina Kauzlaric (Referee) · 25 Mar 2021

**General comments and recommendation**

The manuscript by Meresa et al. presents an interesting illustration of the uncertainties present and arising in modelling flood frequency and flood magnitude under different climate change projections for four catchments in Ireland. While it thoroughly covers a wide range of sources of uncertainty along the modelling chain and their interactions by applying an ANOVA, besides the reduced number of catchments analysed, a main drawback of the setup of the study is the lack of a better representation of the uncertainty stemming from the hydrological modelling. The many different preprocessing methods evaluated and also the multiple extreme value distributions are a strength of this study –which is otherwise not applying nor showing something new-, however the robustness of the finding that hydrological model parameter uncertainty is the least important component is very weak. Furthermore I am surprised the authors (apparently?) didn't expect the results to be different across catchments, as this is the case in some of the studies cited, and not only.

The manuscript is well structured, the methods are generally described in a comprehensible way or supported by relevant sources and/or equations. Even though the discussion provides good points; some important considerations should have been stated already before in the text. While the authors are quite keen on describing findings for each catchment, they don't really try to explain and justify some of the differences observed, what is an important shortcoming.

The manuscript generally features high-quality and interesting figures. However the authors might think about possibly reducing the number of figures (are they all relevant, or could be part of the supplement?), and also improving the readability by changing of some colors used.

I found some inconsistencies in the equations and an error in a figure. I am reporting all those I found in the technical corrections.

Because of these considerations, I think the manuscript requires further work before it can be recommended for publication.

Please find my specific and technical comments here following.

**Specific comments**

- Introduction:
  - This is HESS and not NHESS, but still I think using the term flood risk might be misleading for some readers. I think using flood frequency and flood magnitude would be more appropriate. Avoid also to speak about extremes, as the return periods you are looking at here are those usually considered in many countries as the limiting design floods for inhabitated areas.
  - In general there might be more literature out there to cite, but in particular here I miss Addor et al.2014, who did a similar evaluation for several Swiss river catchments, and have actually some common findings.

- Modelling and numerical experiments:
  - I am not sure numerical experiments is the correct name for what you did.
  - In your paper you make very strong statements about the uncertainty related to the hydrological model parameters, but this might be related to the hydrological model used itself – what you also say later in the discussion- however you might already state this here. I would also expect you to actually better justify your choice: why using a single conceptual hydrological model with only 4 parameters? Why completely leaving out a more physically based model (where the assumption of stationary parameters might be relaxed) ?

- P4-L103: in Table 1 with elevation do you mean *mean* elevation? What is exactly 95% of precipitation value?
- Why do you use only one goodness-of-fit measure –in principle-for selecting the behavioural parameters? Or how do you exactly take PCI into account? This is not clear to me according to your text.
- P8-L230: I think here there is a mistake, GEV has three parameters, whereas Log-Normal and Log-Logistic 2?(again pay attention to the consistency of lower/upper cases and the separating dash between text and Equations)
- Equation 17: you might write $(x-\mu)/\sigma$ instead of $z$ (for the sake of consistency with the other equations).
- P8-L235 $k$ is the shape parameter and not the location parameter, and please define $\sigma$ and $\mu$ too (GEV's scale and location parameters respectively).

- Results:
  - P10 Wouldn't make more sense to have Fig.4 shown and commented before Fig. 3?
  - P10 Fig.6 is a figure you might consider to put in the Supplement
  - P10-L290-295: you could spend a few more words on the deficiency of the model in modelling late summer-autumn, and also explain how the NSE values shown in Fig.7 have been calculated (with daily or monthly data?) It would be also important to spend some few more words also on the performance of the model in the different catchments.
  - Fig.7: you are showing the 95% interval, but this is not in line with your text P10-L293, or I am missing something?
  - P11 first paragraph: I am not sure how "useful" this is. If you want to keep it (and Fig. 8 too), I would suggest you elaborate more on the trend and patterns you are mentioning, on the visible temporary effects, and rather give flow increases as percentages rather than absolute values.
  - P11-L3095-306 isn't there a clear increase with all downscaling methods? Well, RAW data too..
  - P11-L317: you are not showing the reference (i.e. observations) in Fig. 10, so what do you mean by saying ..*the smallest changes in flood quantiles..?*
  - P11-L318: I think here it should be LogN and not LogL?
  - Is the y-axes in Fig.11 correct for the Slaney catchments downscaled using EQM? There is a massive difference as compared to the application of the other downscaling methods, and you don't really mention it in the text…
  - P11-L327: aren't the smallest changes for Slaney using BSM ?
  - P11-L327: aren't the smallest changes for Newport using CF and BSM ?
  - P12-L332-333: why is that? Is this not contradicting results shown in Fig. 5?
  - Figure 13: First of all, some colors are too similar (e.g. BC and BC*DM look almost the same to me), second, it is not so easy to compare the different return periods by eye, as the circles have different diameters, what can be deceptive (e.g. the percentage seems to increase with the return period). It might be helpful to add the actual percentages and write somewhere the return period too. If possible removing the white outline of the percentages might also improve the figure.

- Discussion:
  - P 13-L391: Both Bastola et al. a&b?
  - P14-L399-400: Across all catchments the uncertainty in *future hydrological model parameters* .. is wrong. Please correct resp. reformulate this sentence.

- P15-L453: you might want to add a comment on the influence resp. limitation of assuming flood processes to remain stationary within the 30-year windows on your extreme value distribution fits, and if applying instationary fitting would have been a better option.
- I think authors really need to be harder on themselves for limiting the study to a single specific hydrological model, and elaborate more on what they would expect to be different by applying different model structures.

- Conclusion:
  - An important source of uncertainty in any hydrological setup are the discharge data themselves, which are implicitly assumed to be true resp. correct. There is an interesting study by Westerberg et al.2020 on this topic, it might be added as a source we should start considering too when performing sensitivity and uncertainty propagation studies, as an outlook for future work?

**Technical corrections**

- P5-L138: for the sake of consistency add (EQM)
- P7 Equation 10: remove the square
- P7 Equation 13: check the consistency of lower and upper cases
- P8-L213: remove a bracket in the middle in (Equation(14)
- P8: Equation 14 vs. text=> check the consistency NQi,p vs. NQin,p
- P8 Equation 17 remove ^ in the equation
- Fig. 9: the SSPs have wrong numbers, SSP2 should be SSP3 and ssp3 should be SSP5

References

Addor, N., O. Rössler, N. Köplin, M. Huss, R. Weingartner, and J. Seibert (2014), Robust changes and sources of uncertainty in the projected hydrological regimes of Swiss catchments, Water Resour. Res., 50, 7541–7562, doi:10.1002/ 2014WR015549

Westerberg, I.K., A.E. Sikorska-Senoner, D. Viviroli, D., M. Vis, and J. Seibert (2020), Hydrological model calibration with uncertain discharge data, Hydr. Sci. J.,https://doi.org/10.1080/02626667.2020.1735638

---

## Referee Comment (RC4) · Martina Kauzlaric (Referee) · 25 Mar 2021

I am really sorry for the inconvenience, I completely forgot to report about one important point. please find it as an additional supplement here.

Please also note the supplement to this comment:
https://hess.copernicus.org/preprints/hess-2020-606/hess-2020-606-RC4-supplement.pdf

---

## Author Comment (AC1) · 19 Apr 2021

**Dear Reviewer,**

We appreciate the time and effort that you have dedicated to providing your valuable and constructive feedback on our manuscript. We have considered all comments and outline our response and the changes we propose below. Our response to reviewer comments (black font) are in red font.

**General comments and recommendation**

The manuscript by Meresa et al. presents an interesting illustration of the uncertainties present and arising in modelling flood frequency and flood magnitude under different climate change projections for four catchments in Ireland.

Thank you for this positive comment.

While it thoroughly covers a wide range of sources of uncertainty along the modelling chain and their interactions by applying an ANOVA, besides the reduced number of catchments analysed, a main drawback of the setup of the study is the lack of a better representation of the uncertainty stemming from the hydrological modelling. The many different preprocessing methods evaluated and also the multiple extreme value distributions are a strength of this study –which is otherwise not applying nor showing something new-, however the robustness of the finding that hydrological model parameter uncertainty is the least important component is very weak. Furthermore I am surprised the authors (apparently?) didn't expect the results to be different across catchments, as this is the case in some of the studies cited, and not only.

Thank you for this critique. We have responded to the other reviewers about the catchment sample and the aims of our study. We agree that while thorough, as you identify, a limitation is the lack of consideration of hydrological model structure. Our study is not exhaustive and we will further highlight and discuss the limitations of the study in our revisions. For clarity, we are not surprised that different uncertainties vary across catchments. We do however highlight this finding that the dominant sources of uncertainty can be so different even for a small country like Ireland is novel. The default position in the literature is that climate models are the dominant source. This is even the case in the important paper by Addor et al. (2014) that you direct us to. They highlight that "*While there seems to be a general agreement on the dominant contribution of climate models to the uncertainty in discharge projections, different conclusions were drawn about the contribution of the hydrological models, for example*". Our findings are novel in that they show that uncertainty from bias correction and even extreme value distributions can outweigh the uncertainty in flood quantile projections from a large ensemble of climate models for some catchments. We will be sure to clarify this in our revisions.

We will also integrate further work on parsing the uncertainty in hydrological model parameters, showing how this source of uncertainty is variable, depending on which climate model is selected and the signal of change for different future time periods. For instance initial results are shown in the figure below, whereby the range of change in annual maximum flow from hydrological model parameter uncertainty for 12 CMIP6 climate models are shown for the 2020s, 50s and 80s. This clearly shows that the parameter uncertainty varies depending on the catchment, climate model used and time period considered.

[Figure]

**Figure R1.** Hydrological parameter uncertainty using the selected behavioural parameter sets for the reference period (1976-2005), clim1 (2020s), clim2 (2050s) and clim3 (2080s). The black vertical line and broken horizontal dotted line represent the clim3, the blue vertical line and broken horizontal dot represent the clim2, the green vertical line and broken horizontal dot represent the clim1, and the red vertical line and broken horizontal dot represents the reference period. Each vertical line represents the hydrological parameter uncertainty with its upper and lower values (horizontal). X-axis presents the list of 12 climate models.

You are running 30'000 simulations and applying GLUE, defining as behavioural parameter sets whose simulation show an NSE >0.5. As the only other thing we know that by doing this you, retain 300 parameter sets for you further analysis. I was wondering about the identifiability of parameters and how do their distributions look like in respect to their likelihood. I think these are some important pieces of information you might add to the Supplement material so that the readers are sure how do these calibrated parameters look like, and/or if they are clustering somewhere.

Thank you for the concern. We will add a supplementary figure to the revised manuscript as requested.

The manuscript is well structured, the methods are generally described in a comprehensible way or supported by relevant sources and/or equations. Even though the discussion provides good points; some important considerations should have been stated already before in the text. While the authors are quite keen on describing findings for each catchment, they don't really try to explain and justify some of the differences observed, what is an important shortcoming.

This is a fair point and we will do more to outline differences and reasoning as to why in our revisions.

The manuscript generally features high-quality and interesting figures. However the authors might think about possibly reducing the number of figures (are they all relevant, or could be part of the supplement?), and also improving the readability by changing of some colors used.

Reviewer 2 also recommended this. We will remove Fig 6 and associated text and move Fig 8 to supplementary material.

I found some inconsistencies in the equations and an error in a figure. I am reporting all those I found in the technical corrections.

Thank you for such rigor. We will address all of these issues.

**Specific comments: specific and technical comments here following**

**- Introduction:**

This is HESS and not NHESS, but still I think using the term flood risk might be misleading for some readers. I think using flood frequency and flood magnitude would be more appropriate. Avoid also to speak about extremes, as the return periods you are looking at here are those usually considered in many countries as the limiting design floods for inhabited areas.

Common point across all reviews, we will avoid use of 'risk'. While we agree that they are design floods, we would argue that they are still extremes.

In general there might be more literature out there to cite, but in particular here I miss Addor et al.2014, who did a similar evaluation for several Swiss river catchments, and have actually some common findings.

Thanks for this suggestion, we will integrate all suggested papers. This paper in particular is a very useful suggestion. Addor et al (2014) examined mean flows and highlight the dominance of climate models in catchments, so there are important differences to discuss also, which we will do in revisions.

**- Modelling and numerical experiments:**

I am not sure numerical experiments is the correct name for what you did.

Fair point, we will change to study design.

In your paper you make very strong statements about the uncertainty related to the hydrological model parameters, but this might be related to the hydrological model used itself – what you also say later in the discussion- however you might already state this here. I would also expect you to actually better justify your choice: why using a single conceptual hydrological model with only 4 parameters? Why completely leaving out a more physically based model (where the assumption of stationary parameters might be relaxed) ?

Thanks you for this point. Yes parameter uncertainty may be related to model structure. Our emphasis here was on integration of bias correction and extreme value distributions, together with the new CMIP6 ensemble. We will highlight that our modelling chain is not complete and give further justification for selection of the hydrological model, taking into account studies that have evaluated model structure uncertainty in assessing future high flows. We can add this point that for future researches, considering hydrological models with different degree of complexity, ranging from simple conceptual models to more sophisticated physically-based models can be an interesting subject for researchers.

P4-L103: in Table 1 with elevation do you mean mean elevation? What is exactly 95% of precipitation value?

Yes mean elevation and we will clarify that we refer to the range between the upper and lower ranges of the precipitation distribution. It refers to range of precipitation (mm) between the respective quantiles of the observed data.

Why do you use only one goodness-of-fit measure –in principle-for selecting the behavioural parameters? Or how do you exactly take PCI into account? This is not clear to me according to your text.

We restrict our selection to NSE given our focus on high flows. That said the selection of objective function is another aspect of the uncertainty cascade that may be influential. We will highlight this in our revisions. We will remove the PCI analysis.

P8-L230: I think here there is a mistake, GEV has three parameters, whereas Log-Normal and Log-Logistic 2? (again pay attention to the consistency of lower/upper cases and the separating dash between text and Equations)

Equation 17: you might write $(x-\mu)/\sigma$ instead of $z$ (for the sake of consistency with the other equations).

P8-L235 k is the shape parameter and not the location parameter, and please define $\sigma$ and $\mu$ too (GEV's scale and location parameters respectively).

Thank you. We will correct each of these.

**- Results:**

P10 Wouldn't make more sense to have Fig.4 shown and commented before Fig. 3?

It is possible, however we prefer to leave as is, Fig 3 is a general result and Fig 4 gets into more detail. We feel this is better than vice versa.

P10 Fig.6 is a figure you might consider to put in the Supplement

We will remove this figure

P10-L290-295: you could spend a few more words on the deficiency of the model in modelling late summer-autumn, and also explain how the NSE values shown in Fig.7 have been calculated (with daily or monthly data?) It would be also important to spend some few more words also on the performance of the model in the different catchments.

We can of course do this and tease out how NSE objective function may be an issue for later summer/autumn. We note however that given the timing of floods in Ireland this aspect is less problematic. We will discuss.

Fig.7: you are showing the 95% interval, but this is not in line with your text P10-L293, or I am missing something?

Text should read 95%, we will ensure consistency.

P11 first paragraph: I am not sure how "useful" this is. If you want to keep it (and Fig. 8 too), I would suggest you elaborate more on the trend and patterns you are mentioning, on the visible temporary effects, and rather give flow increases as percentages rather than absolute values.

We will refine and move Fig 8 to supplementary material. We agree this figure is of secondary importance.

P11-L3095-306 isn't there a clear increase with all downscaling methods? Well, RAW data too.

Yes, there is a clear increasing of peak flow in the 2020's, 2050's and 2080's. however, the magnitude is not uniform across the period, bias correction methods, and catchments. Also, it is not a smooth increase from reference period to far future period (2080's).

P11-L317: you are not showing the reference (i.e. observations) in Fig. 10, so what do you mean by saying ..the smallest changes in flood quantiles..?

We are comparing model chains here and will clarify this point.

P11-L318: I think here it should be LogN and not LogL?

Good spot, will correct.

Is the y-axes in Fig.11 correct for the Slaney catchments downscaled using EQM? There is a massive difference as compared to the application of the other downscaling methods, and you don't really mention it in the text…

Thank you for this spot. We will check and revise as necessary.

[Figure]

Revised Figure 11. Percent changes in extreme flow quantiles using ensemble of three distribution types in each catchment for the 2020s (clim1), 2050s (clim2) and 2080s (clim3). Simulated changes are derived using raw (bottom row) and bias corrected (the first four rows) simulations. each line is the mean of changes from 12 climate models and three flood frequency models in the 2020s (red), 2050s (blue) and 2080s (green) period.

P11-L327: aren't the smallest changes for Slaney using BSM ?

It depends on the return period. In the text, we were referencing changes at 20-yrs RT and 100-yrs RT. Nevertheless, you are right if we chose 50-yrs RT. We will clarify the importance of flood magnitude.

P11-L327: aren't the smallest changes for Newport using CF and BSM ?

Again, it depends, see above point and we will clarify.

P12-L332-333: why is that? Is this not contradicting results shown in Fig. 5?

We don't think so but will look further. Fig 5 refers to max annual precipitation. Other aspects are important to changes in flooding and we will expand on this in the revisions.

Figure 13: First of all, some colors are too similar (e.g. BC and BC*DM look almost the same to me), second, it is not so easy to compare the different return periods by eye, as the circles have different diameters, what can be deceptive (e.g. the percentage seems to increase with the return period). It might be helpful to add the actual percentages and write somewhere the return period too. If possible removing the white outline of the percentages might also improve the figure.

Thank you for the comment we will improve presentation in the revised manuscript.

**- Discussion:**

P 13-L391: Both Bastola et al. a&b?

Yes, will update

P14-L399-400: Across all catchments the uncertainty in future hydrological model parameters .. is wrong. Please correct resp. reformulate this sentence.

Will do

P15-L453: you might want to add a comment on the influence resp. limitation of assuming flood processes to remain stationary within the 30-year windows on your extreme value distribution fits, and if applying instationary fitting would have been a better option.

Thank you, this is an important point that we will discuss further, in addition to extrapolating to 100yr flood.

I think authors really need to be harder on themselves for limiting the study to a single specific hydrological model, and elaborate more on what they would expect to be different by applying different model structures.

We agree and will significantly strengthen this aspect.

**- Conclusion:**

An important source of uncertainty in any hydrological setup are the discharge data themselves, which are implicitly assumed to be true resp. correct. There is an interesting study by Westerberg et al.2020 on this topic, it might be added as a source we should start considering too when performing sensitivity and uncertainty propagation studies, as an outlook for future work?

Excellent point, we will integrate this into the discussion also. Thanks!

**Technical corrections**

We will rectify each of the technical corrections. Thanks for being so thorough.

P5-L138: for the sake of consistency add (EQM)

P7 Equation 10: remove the square

P7 Equation 13: check the consistency of lower and upper cases

P8-L213: remove a bracket in the middle in (Equation(14)

P8: Equation 14 vs. text=> check the consistency NQi,p vs. NQin,p

P8 Equation 17 remove ^ in the equation

Fig. 9: the SSPs have wrong numbers, SSP2 should be SSP3 and ssp3 should be SSP5

**References**

Addor, N., O. Rossler, N. K € oplin, M. € Huss, R. Weingartner, and J. Seibert (2014), Robust changes and sources of uncertainty in the projected hydrological regimes of Swiss catchments, Water Resour. Res., 50, 7541–7562, doi:10.1002/ 2014WR015549.

---

## Author Comment (AC4) · 19 Apr 2021

Thank you for the concern. We will add a supplementary figure to the revised manuscript as requested.

---

## Author Response (AR1)

**Dear reviewer,**

We appreciate the time and effort that you have dedicated to providing your valuable and constructive feedback on our manuscript. We have considered all comments and outline the changes we propose below. Below the reviewer comments are in black font and our responses in red.

Reviewer #1

**General comments:**

The manuscript investigates the uncertainty contributions from different components of the modeling chain in future flood magnitude of four Irish catchments. Using ANOVA, the study considers several sources of uncertainty: GCMs, bias-correction, hydrological model parameter, and extreme value distribution and their interactions. Although the manuscript is generally well written and figures are clear, I think it ends up being merely descriptive and is somewhat incomplete as it does not tackle the reasons why uncertainty varies across catchments.

We attempt to link hypotheses as to why the uncertainty partitioning differs between catchments in the discussion section using information, we have about each catchment to infer dominant processes on flood generation. In our revised manuscript we will further develop this discussion as is possible.

We have added further interpretation and results to the manuscript. We feel we have gone as far as we can linking why uncertainties vary across catchments and emphasis this aspect as an objective for future work.

In fact, the main finding I retained after reading the paper was that hydrological model parameter is the least important source of uncertainty (I also don't think there is novelty in stating that sources of uncertainty vary across catchments as emphasized in the conclusions).

We disagree somewhat here. While it is not unsurprising that components of uncertainty differ in catchments it is surprising to us that the dominant source of uncertainty can vary to the extent we show. It is widely communicated in the literature that climate models are the dominant source of uncertainty in impact modelling chains. Each of the studies we cited in the manuscript find this, while Addor et al. (2018), who investigated uncertainty partitioning in mean flows for six Swiss catchments also highlight the dominance of climate models. The point is further illustrated by Giuntoli et al. (2018) who considered four factors (GCMs, GIM, Ivar, RCPs) in partitioning uncertainty of future flow. Their findings confirmed that GCMs are the most dominant factors over the selected study. Lawrence (2020) also compared the uncertainty originated from three factors (CMs, FF models, and hydro pars). Their results confirmed that climate models and extreme frequency models' contribution is similar and followed by the hydrological model parameter. Steinschneider et al., 2014 similarly evaluated the uncertainty from hydrological models, internal climate variability, and hydrological parameters in the observed time series in the US. They concluded that the internal climate variability and hydrologic uncertainty had similar impact on flood risk magnitude. We will reframe this finding to emphasis that the dominant sources of uncertainty change across catchments, even in a small domain like Ireland. We hold this is an important finding, especially for assessment

of flood hazards, whereby the inclusion of multiple extreme value distributions is rarely considered.

We have reframed the emphasis to ensure that the finding that the dominant source of uncertainty can change by catchment and not just the balance of contributions.

As the authors noted in the manuscript, the number of catchments analyzed is small, therefore results do not allow to pinpoint aspects that can further our understanding of why uncertainty shares vary across catchments. The authors themselves write in the final phrase of the conclusions something that I think they should have tackled in their work: "Future work to better understand the link between the key components of the cascade of uncertainty and catchment characteristics is therefore recommended".

Our research questions revolved around examining the partitioning and interaction of uncertainties in future flood magnitude. Blöschl et al., 2019, highlight uncertainty in hydrology as one of the 25 challenges in hydrology science. Our findings highlight that it may (not certain) be possible to a priori identify these features from catchment characteristics. We think it is fair for science to operate in such an incremental way. We agree that in an ideal case we would have many more catchments to more fully interrogate the links between catchment characteristics and the uncertainty cascade. However, we also argue that research is always limited by resources and time. We have used the resource and time available to us to highlight important findings that can be used to inform future work that can more fully unpack these issues and therefore contribute to the science.

No change to paper

**In my opinion, this work could improve through the following:**

The authors could take the advantage of having scrutinized three variables (temperature, precipitation, and runoff) in the control period, using observed data, to select model runs that behaved better, and use them to estimate future magnitudes. This would provide valuable information, because though limited to few catchments it would show the value of forming constrained chains of models and to assess how these compare to the original "big ensemble of runs" in terms of the uncertainty that is associated to return periods of future floods. For instance, once the best bias correction methods are identified (e.g. Figure 4, or L. 431 DGQM, SGQM), these could be used to form a constrained ensemble. This could affect results in uncertainty shares and, importantly, would help reducing the uncertainty.

Thanks for this recommendation. We don't agree that selecting model chains based purely on performance for the observed period 'reduces' uncertainty (e.g. Smith et al. 2020; Clark et al., 2016). This has been shown for bias correction methods (see discussion in paper) and literature on weighting climate model projections (e.g. Weigel et al., 2010). Use of a single 'best' chain does not provide more information to catchment managers and decision managers with respect to their interaction and prediction error (Benke et al., 2008). For Example, using the 95 CI of each factor and their respective interaction, we can provide more information about the role of factors we considered than a single point estimate value using a single approach and informs a single adaptation and mitigation policy about the future flood risk. Rather, as highlighted by Clark et al.,2016, reducing uncertainty is most likely through excluding poor models/methods. We argue that this is very different to selecting the 'best' components of the model chain. We will discuss prospects for reducing uncertainty and sub selecting smaller subsets from large

ensembles to assist decision makers. We also cannot see how we apply the selection of 'best' consistently to the full modelling chain, as this would involve selecting a 'best' climate model.

We have not discussed potential for reducing uncertainty and sub selection. After careful consideration this is a bit of a tangent for our paper and we cannot do so well without taking away from the key message of the paper.

Avoid the word risk (In the title too). What is done in this paper does not deal with flood risk, but rather with changes in future flood magnitude.

Agreed, we will avoid use of risk.

Done

A word of caution on the return periods of 100 years, based on periods of 30 years. Extrapolation of extreme value fits to a domain well outside the range of the reference period (30-yrs) may itself add considerable uncertainty to the estimate. Extreme values are already hard to sample for their rare nature. I would limit return periods to 30yrs to avoid calculation of return values for return periods exceeding the record length. This aspect should be at least discussed.

Thank you for the comment. Yes, at least we need 2/3 data of the return period year (for example, 100 RT from 66 years record data) (Lanxin Hu, et al.,2019). In fact, in this paper, we assumed that the frequency distribution model's extrapolation error is constant for the selected distribution model types. Such assumption is prevalent in many European studies (Kay et al., 2006; Lawrence Debroh, 2020; Meresa and Romanowicz, 2017). We would like to keep our analysis of return periods but will highlight the challenges mentioned in our revised discussion.

Done. We have added to our limitations paragraph in discussion.

Scenarios. Multiple SSP scenarios are employed at one stage, but they are not considered among the different shares of uncertainty (e.g., Fig 13). The authors should justify this choice.

We argue in the text that SSPs can be selected a-priori given the future world that impact assessments are considered for. For instance, studies may want to examine changes in flood magnitude for a more sustainable future, others for a more fossil fuel intensive future. Given this we examine the other uncertainties and their interactions that cannot be decided upon a piori. We will further emphasise this point in the paper and in discussion.

We have justified this choice in the revised paper.

Use of variables: variables are used both at daily and at monthly scales. When dealing with events that have short onset and duration like floods. Possibly, I suggest using the daily time scale at all times.

Yes, we use daily timestep throughout the paper. We see how this was unclear and will clarify in our revisions.

We have clarified use of daily timestep throughout

L. 225 - From this sentence it seems like for a given catchment a single distribution is not sufficient to sample extreme events. I think this is misleading as the practice is to test several

and then choose the one that fits best. In this case you employ multiple distribution types to assess the uncertainty that is brought about by this source of uncertainty.

Yes, we compare different frequency distribution models, and it is possible to choose the one that fits best using AIC. However, it does not mean that it is true across all catchments and climate models and time periods (Table S1). That's why we considered flood frequency distribution models as a very crucial source of uncertainty.

No change to paper.

**Technical corrections**:

We will implement all technical correction listed and thank the reviewer for such detailed commentary.

L. 30 - Would avoid word "significant" (strong statistical connotation).

Done

L. 34 - are shifting -> have shifted

Done

L. 53-57 - I would add Shepherd et al. 2018.

Done

L. 69-71 - It is stated that "an important step [. . .] contribution of different components of uncertainty and their interaction to be quantified and partitioned to help scientists and decision-makers better navigate the cascade of uncertainty". The authors could hint or provide examples on how this information could be used to "better navigate the cascade of uncertainty" and why it is helpful/important.

Done

L. 115-130 - Could this go into the introduction?

Done

L. 230-232 - I would swap order in sentence to respect that of the equations that are listed below.

Done

L. 239-240 - The assessment of future high flows using multiple scenarios has been done, the authors might add Giuntoli et al. 2018.

Done

L. 266 - Why not daily?

Clarified

L. 275-280 - Possible reasons behind the variety of projected changes?

Added further interpretation of reasons

L. 285 - Please add "in the range of" after "to be".

Text removed.

L. 290 - Please justify the choice of 30000 as number of parameter sets.

This is somewhat subjective but the majority of literature uses samples in the range of thousands to tens of thousands to ensure adequate sample of parmater space. We have added justification to the revised manuscript.

L. 338 - Reference to Figure 12: I think showing the median does not add much, it is also hard to see it on the figure.

Figure clarity improved.

L. 375 - "We examined future flood risk". I would rephrase with something like "we examined changes in future flood magnitude". (In line with what is written at line 459).

Done

L. 394-395 - I would attenuate the sensational tone of the sentence.

Done

L. 397 - I counted one.

Done

L 404 - It this stated "results suggest that rather than being the same across catchments" - I don't think anyone has ever presumed that sources of uncertainty remain constant across catchments, and there are studies in the literature showing this already. Please reformulate.

Reframed, this is referring to the dominant source of uncertainty

L. 464-465 - rather than "that the dominant sources of unc. vary on a cathment basis" I suggest writing "our results show how dominant sources of unc. may vary on a catchment basis". I don't find the end of the sentence convincing provided that two adjacent catchments can have entirely different hydrological processes that contribute to flood generation - and in turn different shares of uncertainty sources.

Done

**References:**

Addor, N., O. Rossler, N. K € oplin, M. € Huss, R. Weingartner, and J. Seibert (2014), Robust changes and sources of uncertainty in the projected hydrological regimes of Swiss catchments, Water Resour. Res., 50, 7541–7562, doi:10.1002/ 2014WR015549.

Benke, K., Lowella , K., Hamilton, A.,.Parameter uncertainty, sensitivity analysis and prediction error in a water-balance hydrological model. Mathematical and Computer Modelling 47 (2008) 1134–1149

Blöschi et al., 2019. Twenty-three unsolved problems in hydrology (UPH) – a community perspective. : https://doi.org/10.1080/02626667.2019.1620507

Clark, M. P., Wilby, R. L., Gutmann, E. D., Vano, J. A., Gangopadhyay, S., Wood, A. W., Fowler, H. J., Prudhomme, C., Arnold, J. R., and Brekke, L. D:. Characterizing Uncertainty of the Hydrologic Impacts of Climate Change. Current Climate Change Reports, 2(2), 55–64. https://doi.org/10.1007/s40641-016-0034-x, 2016.

Giuntoli, I., Villarini, G., Prudhomme, C. et al. Uncertainties in projected runoff over the conterminous United States. Climatic Change 150, 149–162 (2018). https://doi.org/10.1007/s10584-018-2280-5

Kay A, Richard G. Jones , Nicholas S. Reynard. RCM rainfall for UK flood frequency estimation. II. Climate change results. Journal of Hydrology 318 (2006) 163–172.

Lawrence, D:. Uncertainty introduced by flood frequency analysis in projections for changes in flood magnitudes under a future climate in Norway. Journal of Hydrology: Regional Studies, 28(December 2019), 100675. https://doi.org/10.1016/j.ejrh.2020.100675, 2020.

Lanxin Hu, Efthymios I. Nikolopoulos,  Francesco Marra, Emmanouil N. Anagnostou. Sensitivity of flood frequency analysis to data record, statistical model, and parameter estimation methods: An evaluation over the contiguous United States. J Flood Risk Management. 2020;13:e12580. https://doi.org/10.1111/jfr3.12580

Meresa, H.K., and Romanowicz, R. J:. The critical role of uncertainty in projections of hydrological extremes. Hydrology and Earth System Sciences, 21(8). https://doi.org/10.5194/hess-21-4245-2017, 2017.

Shepherd, T.G., Boyd, E., Calel, R.A. et al. Storylines: an alternative approach to representing uncertainty in physical aspects of climate change. Climatic Change 151, 555–571 (2018). https://doi.org/10.1007/s10584-018-2317-9

Smith, K. A., Wilby, R. L., Broderick, C., Prudhomme, C., Matthews, T., Harrigan, S., and Murphy, C:. Navigating Cascades of Uncertainty — As Easy as ABC? Not Quite…. Journal of Extreme Events, 05(01), 1850007. https://doi.org/10.1142/s2345737618500070, 2018.

Steinschneider, S., Wi, S., Brown, C., 2015. The integrated effects of climate and hydrologic uncertainty on future flood risk assessments. Hydrol. Process 29, 2823–2839.

Weigel, A.P., Knutti R., Liniger, M.A., Appenzeller, C. (2010) Risks of model weighting in mulimodel climate projections. Journal of Climate, 23(15), 4175-4191

**Dear Reviewer,**

We appreciate the time and effort that you have dedicated to providing your valuable and constructive feedback on our manuscript. We have considered all comments and outline our response and the changes we propose below.

**Reviewer #2**

The paper analyses the propagation of uncertainty in the estimation of future flood quantiles associated with climate change in four Irish catchments. It uses the ANOVA method to quantify the relative importance of different sources of uncertainties, including the choice of the Global Climate Model, the bias correction approach, the parameterisation of the hydrological model to transform climate variables (rainfall and temperature) into flows, and the choice of the extreme value distribution for the flood frequency analysis. The analysis is solid and the paper is clearly written and enjoyable to read.

Many thanks for these positive comments

The main finding is that the dominant sources of uncertainty vary between catchments. This is not a completely new result, as previous studies already showed that different uncertainty sources are dominant in different places (as also noted in the discussion, L. 390-395). In the discussion, the authors attempt at linking some of these differences to catchment characteristics and dominant rainfall-runoff processes in each catchment. However, a comprehensive quantification and evaluation of these links is left to future studies. This is a bit regrettable, as I think the identification of these links is not only the logical next step but also the most challanging and potentially most interesting for future climate change impact assessments - if some of those links could be identified, then one could give a priori indications on which uncertainties should be targeted/reduced first, depending on the characteristics of the catchment under study. This said, I still think the manuscript offers an interesting contribution and is worth publishing after some revisions. Below are some suggestions for improvement.

Thank you. We agree that these are the logical next steps from our findings. As outlined in response to Reviewer 1 we agree that in an ideal case we would require to have much more catchments to more fully interrogate the links between catchment characteristics and the uncertainty cascade. However, we also argue that research is always limited by resources and time. We have used the resource and time available to us to highlight important findings that can be used to inform future work that can more fully unpack these issues and therefore contribute to the science. Our research question aimed at examining the portioning and interaction of uncertainties considered. The link with catchment characteristics was hypothesized, and even though we have a small sample the results suggest that this may be a very fruitful way forward for the reasons you outline. We can unpack additional aspects of how and why doing this might be useful in our revisions. We will also add this point that by linking catchment characteristics to uncertainty sources, it can lead to less effort/time by indicating which uncertainties should be targeted/reduced first.

We have addressed these points in our revision and concluded the paper with this hopeful point of linking catchment characteristics and uncertainties.

The title may be a bit more specific. "Flood risk" may be interpreted as relating to the product of hazard, exposure and vulnerability, whereas this paper only deals with the hazard component. Pereaphs referring to "flood frequency analysis" or "extreme flow magnitudes" in the title would help readers get a better idea of the actual paper content.

We agree and will revise to avoid use of risk.

Done

I think the paper could be made more coincise, in particular the number of figures reduced. I am not sure that all of them are needed to support the key points the authors want to make. Below are some specific suggestions of paragraphs and figures that I think could be removed from the paper and possibly given in Supplementary Material.

Thank you for the suggestions and considered advice.

We have made more concise and removed/moved figures

The Introduction could elaborate a bit more about the ultimate goals of the analysis and previous studies that had similar goals. On Line 69, the authors say "an important step ... is the development of techniques that allow the contribution...". I find this a bit confusing. First, it suggests to me that this paper will deal with "developing" some new technique for sensitivity anaysis, whereas the point here is not to develop techniques but rather use existing ones for a specific application - identify the dominant uncertainties of flood frequency analysis under climate change.

This is a fair point, and we will elaborate the introduction with the valuable suggestions of papers from each reviewer and will also clarity in the aims that we deploy existing methods.

Done

Second, it would be good to discuss here what is to be learnt from such sensitivity analysis, how SA results may inform future climate change impacts assessment. These implications were mentioned in the Abstract, I would have expected to read more on this here too, including the results of previous studies that looked at a similar problem in different catchments (some of these studies are cited in the Discussion on L. 390-395) or studies that used SA to identify dominant uncertainty sources of other hydrological variables. Indeed, that such sources may differ from catchment to catchment has been shown already although pereaphs in different context / for different purposes (see for instance about the variations in dominant parameters of a hydrological model across US cathcments: van Werkhoven et al 2007, Characterization of watershed model behavior across a hydroclimatic gradient, WRR, doi:10.1029/2007WR006271).

Thank you for the point raised. We want to be careful to highlight we do not use sensitivity analysis to identify dominant uncertainty sources. We present here two ways of analyzing uncertainty in future flood hazard, i) an additive chain, which is more similar to sensitivity analysis, and ii) we used ANOVA to separate the variance contribution of each main factor and their interaction. Werkhoven et al., 2008, work is pure sensitivity analysis on hydrological model parameter and hydroclimate regime using SOBOL approach. i.e, our focus is more on

the assessment model outputs uncertainty that derives from an ensemble of four factors. However, we agree that sensitivity analysis helps to understand the contribution of individual factors to uncertainty in output. We will update this aspect of the methodology.

We used two ways of uncertainty analysis in future flood hazard, i) an additive chain, which is more similar to sensitivity analysis. This is mainly applicable in independent driving variables analysis. For example, Werkhoven et al., 2008, used two variables (as one input and the other output), hydrological model parameter and hydroclimate regime, they used SOBOL sensitivity approach to understand the role of inputs (hydrological model parameter) to output (hydroclimate regime), and ii) we used ANOVA to separate the variance contribution of each main factor and their interaction. Unlike the former (sensitivity approach), it helps us to understand the interaction of factors and their main variables. We have added detail to clarify.

I do not understand what the "additive and multiplicative approaches to uncertainty estimation" are and how they work. These are mentioned for the first time on L. 240 ("unlike additive or multiplicative approaches to uncertainty estimation, ANOVA....") but no reference is given. I am familiar with SA literature (for example I know ANOVA) and I would not say it is self-evident what the authors are referring to! The results of this "additive" approach are reported in Fig 12. I am not clear how these resultes (the shaded areas) were obtained (Did you let each uncertainty source vary while keeping all others fixed to their 'reference' value? But if so, how did you choose the reference values? Or maybe you progressively added uncertainty sources one at the time? But if so, the order of adding uncertainty probably conditions your results?). Either way: is this analysis really needed? From a methodological point of view, it seems less robust then the following ANOVA and from a practical point of view it lead to the same conclusions (unless I am missing something?) that the relative importance of the uncertain inputs is catchment dependent. So, does it need reporting?

thank you. It is more related to the third comment. We will add in our methods more detail about additive and multiplicative approaches to uncertainty estimation. The additive approach is the sum of the four selected factors without considering their interaction. The ANOVA is an excellent example of a multiplicative approach, which considered both the variance from the main factor and their respective interaction.

We have added more detail to the manuscript and tried to clarify.

Similarly to previous point, I found the explanation of how PCI works and what it is used for (L.210-220) a bit too concise and hence unclear. On the other hand, I am not really sure mentioning it is really needed, as it does not seem to be used in the Results section 3.2. Please clarify or remove.

Thank you, we will remove this from the manuscript.

We have removed this aspect.

Other figures that the authors may consider moving to Supplementary Materials: Figure 6: is it needed? what for? Indeed text to comment it on L. 282-287 is not particularly informative. Figure 8: again, what's the point? It shows the ensemble median, so it hardly tell us about the

spread of uncertainty, and it brings together past and future behaviour, so it doesn't tell us about the effects of climate change... So, what is the reader expected to learn from it?

We will remove Fig 6 and associated text and move Fig 8 to supplementary material.

Done

The authors discuss this already, but I think it is really worth emphasising even more: not considering structural uncertainty in the hydrological model is a major limitation of this study. Results show that parameter uncertainty has small impact on the final output variability, but really we do not know whether that is because the chosen model structure already condition the range of flow predictions that one can get, and whether maybe those would significantly change with a different model structure. This may not be too problematic if the uncertainty about the model structure was small - in other words, if the authors had a strong justification of why that model structure is particularly adequate for the catchments under study. This could be further elaborated in the revised manuscript.

We agree and will further emphasise this point as a limitation in the study. We will also do more on examining the contribution of parameter uncertainty (see response to Reviewer 3).

We have emphasized this limitation further and indeed others. We have also added new results on parameter uncertainty. We have also referenced previous work to apply GR4J model in Ireland.

Sec. 3.1: I got a bit lost and was not completely sure what the key point is here. It seems to me that, regardless of all details, the key point here is that it is not possible to declare one bias correction method to be the best at improving performances in all catchments, so it remains unclear how to choose one; but the choice matters as the projections for the future change quite a bit depending on the method employed.

Exactly, we will clarify this narrative.

We have clarified by quantifying results further and reflecting this statement. Yes, it is challenging to select one bias correction approach for all the specified catchments. The performance of each bias correction approach is not the same in all the selected catchments. Each method mainly depends on the observed and raw GCMs precipitation characteristics (Teutschbein and Seibert, 2012). Therefore, we compromised by using five bias correction methods and includes their role in future flood hazard, uncertainty.

OTHER SPECIFIC POINTS

We will address/clarify the following technical points.

Squaring in Eq. (10): is it needed? (I would think not)

Done

L. 333: is it sensible that changes are greater in the 2050s than in the 2080s? Isn't it odd?

Not necessarily, it depends on the emissions pathway and on natural climate variability. We use SSP3 which has emissions decreasing in the latter part of the century. This may be due to

the concurrence of climate change physical covariates such as the NAO (North Atlantic Oscillation).

Figure 13: maybe could be made easier to read by inserting the return period in the circle and explaining acronyms in the legend

Done

**Reference**

Werkhoven, K., T. Wagener, P. Reed, and Y. Tang (2008), Characterization of watershed model behavior across a hydroclimatic gradient, Water Resour. Res., 44, W01429, doi:10.1029/2007WR006271.

**Dear Reviewer,**

We appreciate the time and effort that you have dedicated to providing your valuable and constructive feedback on our manuscript. We have considered all comments and outline our response and the changes we propose below. Our response to reviewer comments (black font) are in red font.

**Reviewer #3**

**General comments and recommendation**

The manuscript by Meresa et al. presents an interesting illustration of the uncertainties present and arising in modelling flood frequency and flood magnitude under different climate change projections for four catchments in Ireland.

Thank you for this positive comment.

No change

While it thoroughly covers a wide range of sources of uncertainty along the modelling chain and their interactions by applying an ANOVA, besides the reduced number of catchments analysed, a main drawback of the setup of the study is the lack of a better representation of the uncertainty stemming from the hydrological modelling. The many different preprocessing methods evaluated and also the multiple extreme value distributions are a strength of this study –which is otherwise not applying nor showing something new-, however the robustness of the finding that hydrological model parameter uncertainty is the least important component is very weak. Furthermore I am surprised the authors (apparently?) didn't expect the results to be different across catchments, as this is the case in some of the studies cited, and not only.

Thank you for this critique. We have responded to the other reviewers about the catchment sample and the aims of our study. We agree that while thorough, as you identify, a limitation is the lack of consideration of hydrological model structure. Our study is not exhaustive and we will further highlight and discuss the limitations of the study in our revisions. For clarity, we are not surprised that different uncertainties vary across catchments. We do however highlight this finding that the dominant sources of uncertainty can be so different even for a small country like Ireland is novel. The default position in the literature is that climate models are the dominant source. This is even the case in the important paper by Addor et al. (2014) that you direct us to. They highlight that "*While there seems to be a general agreement on the dominant contribution of climate models to the uncertainty in discharge projections, different conclusions were drawn about the contribution of the hydrological models, for example*". Our findings are novel in that they show that uncertainty from bias correction and even extreme value distributions can outweigh the uncertainty in flood quantile projections from a large ensemble of climate models for some catchments. We will be sure to clarify this in our revisions.

We will also integrate further work on parsing the uncertainty in hydrological model parameters, showing how this source of uncertainty is variable, depending on which climate model is selected and the signal of change for different future time periods. For instance initial results are shown in the figure below, whereby the range of change in annual maximum flow from hydrological model parameter uncertainty for 12 CMIP6 climate models are shown for

the 2020s, 50s and 80s. This clearly shows that the parameter uncertainty varies depending on the catchment, climate model used and time period considered.

[Figure]

**Figure R1.** Hydrological parameter uncertainty using the selected behavioural parameter sets for the reference period (1976-2005), clim1 (2020s), clim2 (2050s) and clim3 (2080s). The black vertical line and broken horizontal dotted line represent the clim3, the blue vertical line and broken horizontal dot represent the clim2, the green vertical line and broken horizontal dot represent the clim1, and the red vertical line and broken horizontal dot represents the reference period. Each vertical line represents the hydrological parameter uncertainty with its upper and lower values (horizontal). X-axis presents the list of 12 climate models.

We have noted the limitations of the study more strongly, including not assessing model structural uncertainty. We have added the additional analysis on model parameter uncertainty to the paper and emphasized the key finding as being that the dominant sources of uncertainty can vary.

You are running 30'000 simulations and applying GLUE, defining as behavioural parameter sets whose simulation show an NSE >0.5. As the only other thing we know that by doing this you, retain 300 parameter sets for you further analysis. I was wondering about the identifiability of parameters and how do their distributions look like in respect to their likelihood. I think these are some important pieces of information you might add to the Supplement material so that the readers are sure how do these calibrated parameters look like, and/or if they are clustering somewhere.

Thank you for the concern. We will add a supplementary figure to the revised manuscript as requested.

Done

The manuscript is well structured, the methods are generally described in a comprehensible way or supported by relevant sources and/or equations. Even though the discussion provides good points; some important considerations should have been stated already before in the text.

While the authors are quite keen on describing findings for each catchment, they don't really try to explain and justify some of the differences observed, what is an important shortcoming.

This is a fair point and we will do more to outline differences and reasoning as to why in our revisions.

Done

The manuscript generally features high-quality and interesting figures. However the authors might think about possibly reducing the number of figures (are they all relevant, or could be part of the supplement?), and also improving the readability by changing of some colors used.

Reviewer 2 also recommended this. We will remove Fig 6 and associated text and move Fig 8 to supplementary material.

Done

I found some inconsistencies in the equations and an error in a figure. I am reporting all those I found in the technical corrections.

Thank you for such rigor. We will address all of these issues.

Done

**Specific comments: specific and technical comments here following**

**- Introduction:**

This is HESS and not NHESS, but still I think using the term flood risk might be misleading for some readers. I think using flood frequency and flood magnitude would be more appropriate. Avoid also to speak about extremes, as the return periods you are looking at here are those usually considered in many countries as the limiting design floods for inhabitated areas.

Common point across all reviews, we will avoid use of 'risk'. While we agree that they are design floods, we would argue that they are still extremes.

Done

In general there might be more literature out there to cite, but in particular here I miss Addor et al.2014, who did a similar evaluation for several Swiss river catchments, and have actually some common findings.

Thanks for this suggestion, we will integrate all suggested papers. This paper in particular is a very useful suggestion. Addor et al (2014) examined mean flows and highlight the dominance of climate models in catchments, so there are important differences to discuss also, which we will do in revisions.

Done

**- Modelling and numerical experiments:**

I am not sure numerical experiments is the correct name for what you did.

Fair point, we will change to study design.

Done

In your paper you make very strong statements about the uncertainty related to the hydrological model parameters, but this might be related to the hydrological model used itself – what you also say later in the discussion- however you might already state this here. I would also expect you to actually better justify your choice: why using a single conceptual hydrological model with only 4 parameters? Why completely leaving out a more physically based model (where the assumption of stationary parameters might be relaxed) ?

Thanks you for this point. Yes parameter uncertainty may be related to model structure. Our emphasis here was on integration of bias correction and extreme value distributions, together with the new CMIP6 ensemble. We will highlight that our modelling chain is not complete and give further justification for selection of the hydrological model, taking into account studies that have evaluated model structure uncertainty in assessing future high flows. We can add this point that for future researches, considering hydrological models with different degree of complexity, ranging from simple conceptual models to more sophisticated physically-based models can be an interesting subject for researchers.

Done

P4-L103: in Table 1 with elevation do you mean mean elevation? What is exactly 95% of precipitation value?

Yes, mean elevation.  The 95% of precipitation value is excluded from the Table 1.

Done

Why do you use only one goodness-of-fit measure –in principle-for selecting the behavioural parameters? Or how do you exactly take PCI into account? This is not clear to me according to your text.

We restrict our selection to NSE given our focus on high flows. That said the selection of objective function is another aspect of the uncertainty cascade that may be influential. We will highlight this in our revisions. We will remove the PCI analysis.

Done

P8-L230: I think here there is a mistake, GEV has three parameters, whereas Log-Normal and Log-Logistic 2? (again pay attention to the consistency of lower/upper cases and the separating dash between text and Equations)

Equation 17: you might write $(x-\mu)/\sigma$ instead of $z$ (for the sake of consistency with the other equations).

P8-L235 k is the shape parameter and not the location parameter, and please define $\sigma$ and $\mu$ too (GEV's scale and location parameters respectively).

Thank you. We will correct each of these.

Done

**- Results:**

P10 Wouldn't make more sense to have Fig.4 shown and commented before Fig. 3?

It is possible, however we prefer to leave as is, Fig 3 is a general result and Fig 4 gets into more detail. We feel this is better than vice versa.

No change

P10 Fig.6 is a figure you might consider to put in the Supplement

We will remove this figure

Done

P10-L290-295: you could spend a few more words on the deficiency of the model in modelling late summer-autumn, and also explain how the NSE values shown in Fig.7 have been calculated (with daily or monthly data?) It would be also important to spend some few more words also on the performance of the model in the different catchments.

We can of course do this and tease out how NSE objective function may be an issue for later summer/autumn. We note however that given the timing of floods in Ireland this aspect is less problematic. We will discuss.

Done

Fig.7: you are showing the 95% interval, but this is not in line with your text P10-L293, or I am missing something?

Text should read 95%, we will ensure consistency.

Done

P11 first paragraph: I am not sure how "useful" this is. If you want to keep it (and Fig. 8 too), I would suggest you elaborate more on the trend and patterns you are mentioning, on the visible temporary effects, and rather give flow increases as percentages rather than absolute values.

We will refine and move Fig 8 to supplementary material. We agree this figure is of secondary importance.

Removed from paper

P11-L3095-306 isn't there a clear increase with all downscaling methods? Well, RAW data too.

Yes, there is a clear increasing of peak flow in the 2020's, 2050's and 2080's. however, the magnitude is not uniform across the period, bias correction methods, and catchments. Also, it is not a smooth increase from reference period to far future period (2080's). Especially, the spread is not the same across the bias correction methods. EQM shows wider spread than the other.

P11-L317: you are not showing the reference (i.e. observations) in Fig. 10, so what do you mean by saying ..the smallest changes in flood quantiles..?

We are comparing model chains here and will clarify this point.

It is not possible to show the reference (i.e. observations) in Fig. 10, because, we used 130 years of projections to calculate flood quantiles using different frequency distribution models (GEV, LogN, LogL) at different return periods. Whereas, the observed time series has 30 years. So, not right to compare quantiles derived from 130 yrs and 30yrs. Rather the point in this figure is to compare the 95% confidence interval between different components of the model chain via the additive chain approach. This is now Figure 9 in the revised manuscript.

P11-L318: I think here it should be LogN and not LogL?

Good spot, will correct.

Done

Is the y-axes in Fig.11 correct for the Slaney catchments downscaled using EQM? There is a massive difference as compared to the application of the other downscaling methods, and you don't really mention it in the text…

Thank you for this spot. We will check and revise as necessary.

We have revised this figure which contained a mistake.

P11-L327: aren't the smallest changes for Slaney using BSM ?

It depends on the return period. In the text, we were referencing changes at 20-yrs RT and 100-yrs RT. Nevertheless, you are right if we chose 50-yrs RT. We will clarify the importance of flood magnitude.

Done

P11-L327: aren't the smallest changes for Newport using CF and BSM ?

Again, it depends, see above point and we will clarify.

Done

P12-L332-333: why is that? Is this not contradicting results shown in Fig. 5?

We don't think so but will look further. Fig 5 refers to max annual precipitation. Other aspects are important to changes in flooding and we will expand on this in the revisions.

We don't think so, b/c all extreme precipitations do not necessary produce peak flow (Sharma, et al., 2018). Mostly it depends on the duration and intensity of precipitation.

No change.

Figure 13: First of all, some colors are too similar (e.g. BC and BC*DM look almost the same to me), second, it is not so easy to compare the different return periods by eye, as the circles have different diameters, what can be deceptive (e.g. the percentage seems to increase with the return period). It might be helpful to add the actual percentages and write somewhere the return period too. If possible removing the white outline of the percentages might also improve the figure.

Thank you for the comment we will improve presentation in the revised manuscript.

Done

**- Discussion:**

P 13-L391: Both Bastola et al. a&b?

Yes, will update

Done

P14-L399-400: Across all catchments the uncertainty in future hydrological model parameters .. is wrong. Please correct resp. reformulate this sentence.

Will do

Done

P15-L453: you might want to add a comment on the influence resp. limitation of assuming flood processes to remain stationary within the 30-year windows on your extreme value distribution fits, and if applying instationary fitting would have been a better option.

Thank you, this is an important point that we will discuss further, in addition to extrapolating to 100yr flood.

Done

I think authors really need to be harder on themselves for limiting the study to a single specific hydrological model, and elaborate more on what they would expect to be different by applying different model structures.

We agree and will significantly strengthen this aspect.

Done

**- Conclusion:**

An important source of uncertainty in any hydrological setup are the discharge data themselves, which are implicitly assumed to be true resp. correct. There is an interesting study by Westerberg et al.2020 on this topic, it might be added as a source we should start considering too when performing sensitivity and uncertainty propagation studies, as an outlook for future work?

Excellent point, we will integrate this into the discussion also. Thanks!

Done

**Technical corrections**

We will rectify each of the technical corrections. Thanks for being so thorough.

P5-L138: for the sake of consistency add (EQM)

Done

P7 Equation 10: remove the square

Done

P7 Equation 13: check the consistency of lower and upper cases

Done

P8-L213: remove a bracket in the middle in (Equation(14)

Done

P8: Equation 14 vs. text=> check the consistency $NQ_{i,p}$ vs. $NQ_{in,p}$

Done

P8 Equation 17 remove ^ in the equation

Done

Fig. 9: the SSPs have wrong numbers, SSP2 should be SSP3 and ssp3 should be SSP5

Done

**References**

Addor, N., O. Rossler, N. K € oplin, M. € Huss, R. Weingartner, and J. Seibert (2014), Robust changes and sources of uncertainty in the projected hydrological regimes of Swiss catchments, Water Resour. Res., 50, 7541–7562, doi:10.1002/ 2014WR015549.

---

## Referee Report (RR1)

I thank the authors for replying and tackling most of the comments and corrections of the first review round.
The manuscript has improved and matured, I still have some minor comments and corrections, I report these here following.

Content corrections:

-P 8-L233-235: If I understand correctly the only way you used the 95% confidence interval (PCI) is shown in Fig.6 to evaluate the performance of the hydrological model. If it is so, the second sentence (This is helpful for identifying..) is disturbing, as you didn't use it for selecting the NSE threshold..?or you did? If it is so, please state it more clearly.

- Subsection 2.6.1: Either clarify and lead better the reader through the equations, or rather cite literature where this is already better resp. thoroughly exemplified (see e.g. Bosshard et al.2013) For example say what is q, use a clearer notation, etc..

- P12-L344. ..in particular in the Newport catchment..? I don't see it as particularly identifiable in the other catchments either.

-Subsection 3.3:
- P13-L379-382: here a couple of things went wrong..I guess the sentence should be: *Overall, the LN distribution returned the smallest flood quantile magnitude, while the estimated quantile values using LogL tend to be largest across each catchment, while having the narrowest uncertainty band*..?
- In Fig. 10 sometimes the changes depending on the period (clim1,2,3) get "switched". E.g. respect to the RAW data in Newport, where the smallest changes would be far in the future, but then become the largest e.g. applying BSM. I know it's not a linear system, but could you comment on these switches and where do you think these come from?

Technical corrections:
- P2-L53: there is one parentheses too much before Giuntoli et al.
-P2-L58: there is a space missing after Jobst et al.(2018)
-P3-L1: ..their controbutions tend to vary depending on..
-Table 1: I think you should provide the units everywhere?
- Eq. (16) on the side K should be low case
-P15-L461: remove the bracket in front of Addor, and add a space after ,2014)

Bosshard, T., M. Carambia, K. Goergen, S. Kotlarski, P. Krahe, M. Zappa, and C. Schär (2013), Quantifying uncertainty sources in an ensemble of hydrological climate-impact projections, Water Resour. Res., 49, 1523–1536, doi:10.1029/2011WR011533.

---

## Author Response (AR2)

**Dear editor and reviewers**

Thank you for the additional comments and suggestions.

**Reviewer #1**

RC1. I thank the authors for replying and tackling most of the comments and corrections of the first review round. The manuscript has improved and matured,
AR1. Thank you.

I still have some minor comments and corrections, I report these here following.
Content corrections:

RC2. -P 8-L233-235: If I understand correctly the only way you used the 95% confidence interval (PCI) is shown in Fig.6 to evaluate the performance of the hydrological model. If it is so, the second sentence (This is helpful for identifying..) is disturbing, as you didn't use it for selecting the NSE threshold..? or you did? If it is so, please state it more clearly.
AR2. Yes, we used the PCI for selecting best parameter sets for understanding of hydrological parameter uncertainty. Figure 6. to evaluate the 95% CI performance of the hydrological model. Yes, it is helpful for identifying / selecting best NSE threshold.

RC3- Subsection 2.6.1: Either clarify and lead better the reader through the equations, or rather cite literature where this is already better resp. thoroughly exemplified (see e.g. Bosshard et al.2013) For example say what is q, use a clearer notation, etc..
AR3. Subsection 2.6.1, now we add the description of notations. Thank you for the recommendation to include Bosshard et al.2013 paper. However, this paper is more related to subsection 2.6.2. and it is included.

RC4- P12-L344. ..in particular in the Newport catchment..? I don't see it as particularly identifiable in the other catchments either.
AR4. we mean that the X1 parameter in Newport catchment does not show shape / converge point in to one direction as the performance of the model increases. This is particularly evident in the Newport catchment and we highlight this. The parameter is better defined in other catchments.

-Subsection 3.3:
RC5. • P13-L379-382: here a couple of things went wrong..I guess the sentence should be: Overall, the LN distribution returned the smallest flood quantile magnitude, while the estimated quantile values using LogL tend to be largest across each catchment, while having the narrowest uncertainty band..?
AR5. We have edited as suggested.

AR6. • In Fig. 10 sometimes the changes depending on the period (clim1,2,3) get "switched". E.g. respect to the RAW data in Newport, where the smallest changes would be far in the future, but then become the largest e.g. applying BSM. I know it's not a linear system, but could you comment on these switches and where do you think these come from?
AR6. Yes, the changes depend on the climate period. We added
Interestingly, the changes depend on the future climate period withmost catchments showing higher change in the 2020s than the 2050s. This is likely due to climate variability and nonlinearity climate response.

RC7. Technical corrections:
 - P2-L53: there is one parentheses too much before Giuntoli et al.
-P2-L58: there is a space missing after Jobst et al.(2018)
-P3-L1: ..their controbutions tend to vary depending on..
-Table 1: I think you should provide the units everywhere? - Eq. (16) on the side K should be low case
-P15-L461: remove the bracket in front of Addor, and add a space after ,2014)
AR7. Corrected

Bosshard, T., M. Carambia, K. Goergen, S. Kotlarski, P. Krahe, M. Zappa, and C. Schär (2013), Quantifying uncertainty sources in an ensemble of hydrological climate-impact projections, Water Resour. Res., 49, 1523–1536, doi:10.1029/2011WR011533

**Reviewer #2**

RC1. I think the authors addressed the concerns raised in the first round of review and the manuscript is now acceptable for publication.
AR1. Thank you.

Few small points that could be revised in the final version are listed below.
RC 2. L. 44: "estimates of future greenhouse..." should be "scenarios of future greenhouse ..."
AR2. Done

RC 3. L. 65 "depend on" should be "depending on"
AR3. Done

RC4. L. 82-87 bit confusing. Specifically: " which components of the modelling chain to include for specific catchments." does "components" mean "model component" or (as I guess) source of uncertainty? L. 85 "is forced to choose which aspects of uncertainty to prioritise" I am not sure I understand what the authors mean here. I think there is no need to prioritise uncertainties in a scenario-neutral / bottom up approach. At most I would say the modeller must choose "which sources of uncertainty to include"
AR4. Have clarified

RC5. L. 263-263 "which is similar to sensitivity analysis" ... "Unlike the former (sensitivity approach)," I find this use of the term "sensitivity analysis" a bit confusing (to me, sensitivity analysis encompasses one-at-the time, local and global approaches - and I would say ANOVA could be regarded as a sensitivity analysis, of the "global" kind!). Maybe saying "one-at-the-time sensitivity analysis" here would be more precise?
AR5. Yes, it is true if we use different combination of input variables at a given time. However, in our case, we used ANOVA on the output of our cascade uncertainty model. We have added the clarity requested.

RC 6. L. 404-406: "Therefore, for the spread for climate models was evaluated by varying simulations using each of the 12 GCMs while using a single bias correction technique, median hydrological model parameter set and a single flood frequency distribution" Something strange with this sentence, please revise.
AR4. Done